

# Sensitivity analysis of numerical modeling input parameters on floating offshore wind turbine loads in extreme idling conditions

Will Wiley[1], Jason Jonkman[1], and Amy Robertson[1]

[1]National Renewable Energy Laboratory, 15013 Denver West Parkway, Golden, CO 80401, USA

**Correspondence:** Will Wiley (will.wiley@nrel.gov)

**Abstract.**

Floating offshore wind turbine systems are subject to complex environmental loads, with significant potential for damage in extreme storm conditions. Design simulations in these conditions are required to assess the survivability of the device with some level of confidence. Aero-hydro-servo-elastic engineering tools can be used with a reasonable balance of accuracy and

5 computational efficiency. The models require many input parameters to describe the air and water conditions, the system properties, and the load calculations. Each of these parameters has some possible range, either due to statistical uncertainty or variations with time. Variation in the input parameters can have important effects on the uncertainty in the resulting loads, but it is not practical to perform detailed assessments of the impact of this uncertainty for every input parameter. This work demonstrates a method to identify the input parameters that have the most impact on the loads to focus further inspection.

The process is done specifically for extreme storm load cases defined in the International Electrotechnical Commission design requirements for floating offshore wind turbines. The analysis was performed using the International Energy Agency Wind Technology Collaboration Programme 15 MW offshore reference wind turbine atop the University of Maine VolturnUS-S reference platform in two U.S. offshore wind regions, the Gulf of Maine and Humboldt Bay. It was found that the direction of incident waves and current, yaw misalignment, and the length of mooring line sections were among the primary sensitivities.

## 1 Introduction

Floating offshore wind turbines are complex systems governed by many coupled physical effects. Numerical models of these devices are critical for understanding their performance and loads and the associated risks. Computationally efficient models are needed to optimize design and simulate large sets of design load cases. Aero-hydro-servo-elastic time domain engineering-fidelity tools, such as OpenFAST, are commonly used for these types of simulations (NREL, 2024). These tools require a large

number of input variables that describe the incident wind; aerodynamic loads; system geometry, mass/inertia, and stiffness; controller response; incident wave and current environment; and hydrodynamic loads. There are potentially thousands of variable inputs, and all of these inputs have some associated range of variability. This range is an aggregate of various forms of uncertainty and changes over the life of a project. Changes to the input parameters can significantly influence the resulting ultimate and fatigue design loads. Thorough uncertainty analyses are not feasible for all input parameters but should be performed

for variables that the relevant loads are most sensitive to.





An elementary effects (EE) approach has been developed and demonstrated to identify the parameters creating the largest sensitivities. Demonstrative case studies have been reported that investigate incident wind parameters (Robertson et al., 2018); airfoil properties (Shaler et al., 2019); incident wind and structural property parameters (Robertson et al., 2019); wind, structure, and wake parameters for a small wind farm (Shaler et al., 2021); wind, wave, and turbine parameters for the fatigue of monopile-

supported offshore turbines (Sørum et al., 2022); wind, wave, and structure parameters for a floating offshore wind turbine (Wiley et al., 2023); and wind and wave parameters for two types of floating wind turbine platforms (Reddy et al., 2024). Each of these previous studies was performed with operational load cases with mean wind speeds between cut-in and cut-out speeds. Above-cut-out wind speeds do not occur frequently, but they can potentially cause design-driving ultimate and fatigue loads. The International Electrotechnical Commission (IEC) recognizes the importance of this possibility with the inclusion of design

load cases 6.1–6.5 for parked or idling turbines (IEC, 2024). It is also expected that the most dominant sensitivities may be different in these extreme idling conditions compared to when a turbine is operating.

A previous study of the load sensitivities of a floating offshore wind platform found very limited sensitivity to wave parameters for ultimate loads during normal operation and only secondary sensitivity to wave parameters for fatigue loads. The primary input parameter driving both types of loads was found to be the turbulence intensity of the incident wind (Wiley et al.,

2023). With a nonoperational turbine in extreme conditions, it is expected that this relative significance could change.

This work uses the EE technique to identify the input parameter ranges that need the most attention when considering ultimate and fatigue loads in extreme idling conditions for a 15 MW reference floating wind system.

## 2    Approach

The previously developed and demonstrated EE analysis method was implemented in this work using OpenFAST to model the

International Energy Agency Wind Technology Collaboration Programme (IEA Wind) 15 MW offshore reference wind turbine (RWT) with the University of Maine (UMaine) VolturnUS-S reference platform.

### 2.1    Load Cases

Four load cases were selected from the IEC floating offshore wind design requirement technical specification (IEC, 2024). There are five recommended idling load cases (LC 6.1–6.5) without fault conditions; one of these (6.2) highlights a loss of

electrical network and is not commonly used by industry due to the prevalence of battery backup systems; therefore, it is not used in this work either. Three of the remaining load cases (6.1, 6.3, and 6.5) assess ultimate loads, and one of the load cases (6.4) assesses fatigue loads. The load cases are described at a high level in Table 1. The ultimate load cases feature extreme wind, wave, and current conditions with return periods of 50 years, 1 year, and 500 years, respectively. The 1-year return period of LC 6.3 also includes an extreme range of yaw misalignment. The 500-year return period of LC 6.5 is a robustness check

with a safety factor of 1.0 (compared to a typical ultimate value of 1.35) and is only meant to be checked for global system survival. Fault conditions are not considered in this work.





**Table 1.** Selected IEC design load cases

| IEC DLC | Wind Speed | Significant Wave Height | Current Speed | Other Conditions | Load Type | Partial Safety Factor |
|---|---|---|---|---|---|---|
| **6.1** | 50 yr | 50 yr | 50 yr | | Ultimate | 1.35 |
| **6.3** | 1 yr | 1 yr | 1 yr | Extreme Yaw Misalignment | Ultimate | 1.35 |
| **6.4** | [cut-out , $0.7 \times 50$ yr] | Joint Probability Distribution | Normal | | Fatigue | 1.00 |
| **6.5** | 500 yr | 500 yr | 500 yr | | Ultimate | 1.00 |

Each of the load cases was studied for two locations: the Gulf of Maine on the U.S. Atlantic coast and Humboldt Bay on the U.S. Pacific coast. These two areas have the potential to be two of the first sites for floating offshore wind development in the United States. There is also publicly available metocean data near these locations. Efforts have been made to characterize these sites both for deployments and for use as academic references (Biglu et al., 2024).

## 2.2 System

Offshore wind turbines have continued to increase in size in recent years, with larger diameters and more flexible blades. In 2020, a team from the National Renewable Energy Laboratory (NREL), the Technical University of Denmark, and the University of Maine published the specifications for a 15 MW offshore reference turbine through IEA Wind Task 37. This turbine was designed to represent the anticipated near-future of offshore turbine deployment (Gaertner et al., 2020).

IEA Wind Task 37 also included the design of a reference floating platform to support the 15 MW turbine. The specifications of the semisubmersible UMaine VolturnUS-S were published in 2020 by a team from the University of Maine and NREL. The design features a center-mounted tower with a three-spoke base made of fully submerged rectangular horizontal members and vertical surface-piercing cylinders (Allen et al., 2020). The reference turbine and platform are shown in Fig. 1 with some global dimensions.

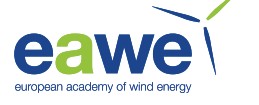


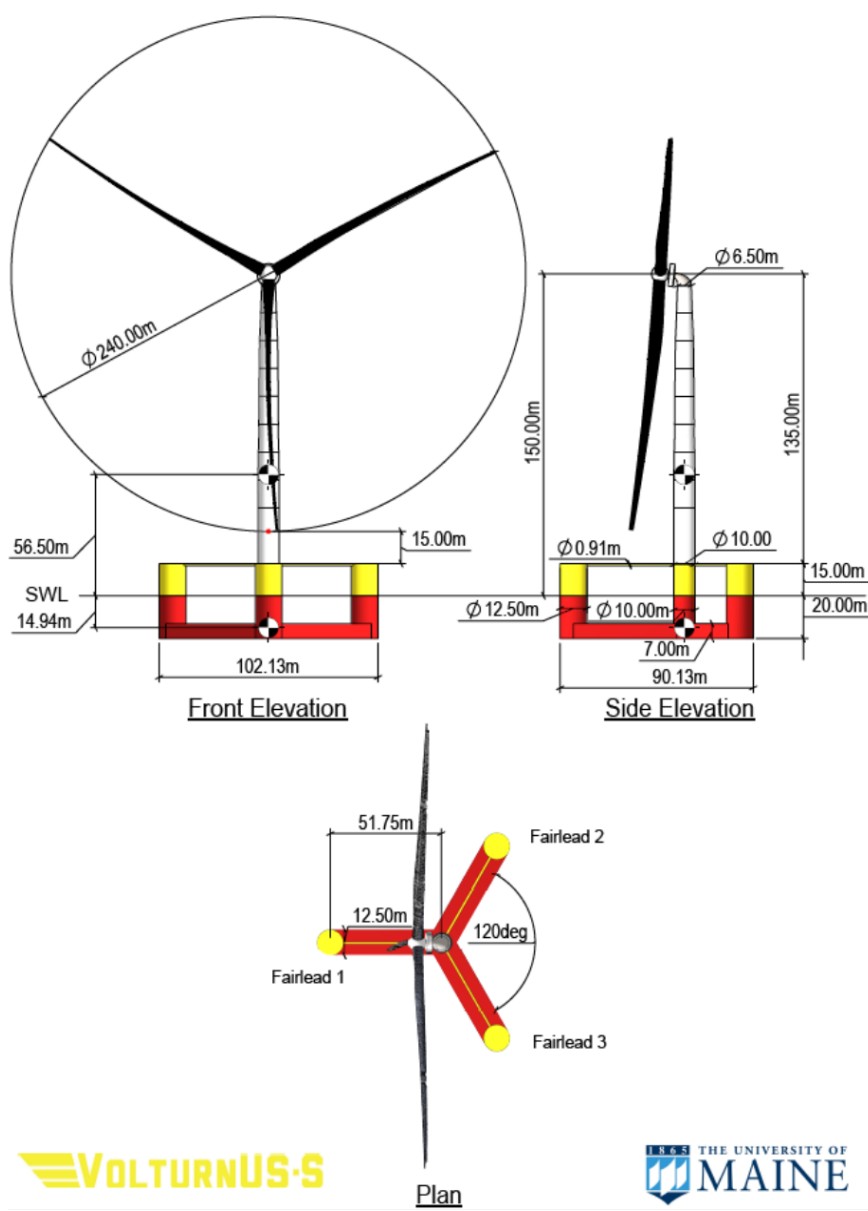

**Figure 1.** UMaine VolturnUS-S reference platform with the IEA 15 MW offshore reference wind turbine (Allen et al., 2020)

### 2.2.1 Mooring Systems

The original design of the UMaine VolturnUS-S included a chain catenary mooring system in 200 m deep water. This water
depth is within the range found in the Gulf of Maine offshore wind area, but shallower than the range found in the Humboldt
Bay offshore wind area, which is between 550 m and 1000 m deep (Cooperman et al., 2022). A team from NREL designed
alternative reference mooring systems for this system in three locations, including the Gulf of Maine and Humboldt Bay. Both





systems include sections of polyester fiber and chain. The design for the Gulf of Maine is an intermediate-depth semi-taut system, and the design for Humboldt Bay is a deep-water taut system. The systems were optimized using a quasi-static tool and verified with dynamic simulations (Lozon et al., 2024). These reference mooring designs are used in this analysis, and visualizations of the systems from the mooring specification publication are shown in Fig. 2.

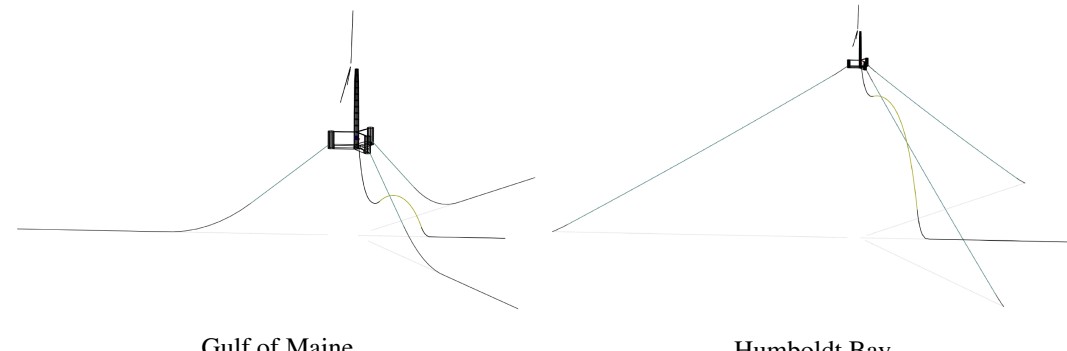

Gulf of Maine                                          Humboldt Bay

**Figure 2.** Mooring system visualizations (Lozon et al., 2024) CAN'T INCLUDE THIS YET, NEED TO SEE IF THE MOORING PAPER HAS BEEN SUBMITTED ALREADY

## 2.3 Modeling

The open-source coupled aero-hydro-servo-elastic tool OpenFAST version 4.0 was used for the numerical modeling (NREL, 2024). The incident turbulent wind field was generated using TurbSim. The AeroDyn version 15 module was used for aerodynamic loads. No induction model was calculated, given the idling rotor.

Unsteady airfoil aerodynamics were not used either. In an idling condition, especially with significant angles of yaw misalignment, the angles of attack on the airfoils can change rapidly and with large magnitudes. Dynamic stall models are intended to capture the unsteady effects on airfoil loading due to changes in the relative inflow but are designed and tuned for angles of attack at or below stall. The deep stall potentially experienced in the load cases of this study is not well captured by existing dynamic stall models. A group from DNV demonstrated in 2023 the dependency of blade load predictions on the choice of dynamic stall model for idling conditions. They compared predictions made with two different dynamic stall models to predictions made with steady aerodynamics using the IEA 15 MW turbine and found that loads and resulting deflections could change by an order of magnitude with a yaw misalignment of $20°$. Mean values were similar in the comparison, but the amplitude of load oscillation was much larger when no dynamic stall model was used (Bangga et al., 2023). Dynamic stall models allow the lift coefficient to temporarily exceed the steady maximum before a sharper collapse, potentially resulting in increased aerodynamic damping. Existing dynamic stall models are not able to accurately represent the large angles of attack expected to occur in this study; therefore, instead of using a correction not suited for the application, only steady airfoil aerodynamics (based on geometric angle of attack) were used (Fuchs et al., 2023). Future work should be done to improve dynamic stall models for the relevant conditions.



The air environment was modeled with the InflowWind module with turbulent inflow derived from TurbSim. A user-defined wind profile was used with a power law shear profile and linear veer profile around the value at the hub height. A Kaimal spectrum and spatial coherence model similar to previous analyses were used, which focus on parameters in the mean wind direction (Robertson et al., 2018).

The water environment was modeled with the SeaState module. Irregular waves were used with an embedded constrained maximum wave crest and first-order kinematics with a superimposed steady current. The hydrodynamic forces were calculated with the HydroDyn module. The platform was represented with a hybrid model, whereby the first-order and second-order difference-frequency potential-flow coefficients were calculated using WAMIT. Strip theory transverse and axial drag coefficients were used for viscous loading.

The ElastoDyn module was used to model blade and tower deflections with predefined mode shapes. There are known inaccuracies in the reduced-order Euler–Bernoulli beam model for large deformations in flexible blades. ElastoDyn was selected over the more accurate geometrically exact beam model in the BeamDyn module due to computational cost, which will be improved through tight coupling in the upcoming OpenFAST version 5.0. The EE analysis requires a large number of simulations to be run, and the increased computational cost was not considered manageable before version 5.0 is available. Structural damping is also difficult to accurately capture, especially for large deflections (Chetan and Bortolotti, 2024). To avoid non-physical blade aeroelastic instabilities at high angles of yaw misalignment, exaggerated by potentially missing aerodynamic damping from the unmodeled dynamic stall, a variable structural damping value was used for the blades. A value of 1% of critical was used when the magnitude of yaw misalignment was below $10°$; above $10°$, the damping linearly increased to 2% of critical at a yaw misalignment magnitude of $20°$. The floating platform was treated as fully rigid to avoid high computational expense.

The mooring system was modeled using the lumped-mass-dynamics-based MoorDyn module.

Each simulation was run for 3800 s, resulting in a 1 h time series after a 200 s transient was removed. This transient period was determined from initial simulations with nominal input parameter values.

## 2.4 Elementary Effects

The EE method was defined by Max Morris in 1991 for general computational experiments. The technique does not identify coupling between parameters, but it reduces the total number of required simulations for a sensitivity assessment (Morris, 1991). Robertson et al. (2018) used a modification of the method that uses radial perturbations of all parameters for a sufficiently large number of starting points. This modified method has been demonstrated in previous studies and is applied to this analysis for an idling turbine in above-cut-out conditions. The approach is represented for a simplified three-parameter system in Fig. 3. The blue points represent starting points in the parameter hyperspace, and the red points are a small perturbation from the starting point in one parameter dimension only. The effect on a selected output quantity of interest (QOI) from the perturbation can be used to calculate a form of a local partial derivative scaled by the range of the parameter. Comparisons of these values calculated with different input parameter perturbations indicate the relative sensitivity to each input at that point





in the hyperspace. When a sufficiently large number of starting points are used, the resulting sensitivities converge toward the global sensitivity values. In this work, the parameter hyperspace included 40 parameters.

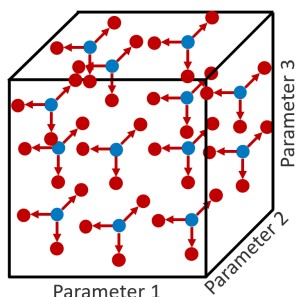

**Figure 3.** Three-parameter representation of input parameter hyperspace with a set of starting points shown in blue and individual parameter perturbation shown in red, adapted from Shaler et al. (2019)

The magnitude of the perturbations was held to a constant value of $\pm 10\%$ of the parameter range. The direction of the perturbation was randomly chosen with the constraint that the perturbed parameter still needed to be within the defined range. The starting point locations followed a Sobol sequence, which leads to a uniform distribution where added points evenly fill the parameter hyperspace, starting with the hyperspace midpoint (Sobol, 1967).

Each test point was run with a number of random seed numbers used for the irregular wave and turbulent wind field generation, and the convergence of the sensitivities was tracked to ensure that a sufficient number of seed numbers was used. Both the ultimate and the fatigue loads were averaged across all seed numbers for a given set of inputs. Ultimate output loads were used for LC 6.1, 6.3, and 6.5 and were calculated following Eq. (1), where $Y$ is an output QOI, $U$ is a certain set of inputs, and $S$ is the number of random seed numbers. $PSF$ is the partial safety factor as designated by load case in Table 1.

$$Y_{ult.}(U) = \frac{PSF}{S} \sum_{s=1}^{S} MAX(|Y(U)|) \tag{1}$$

Fatigue loads were used for LC 6.4 and were calculated as damage-equivalent loads (DELs) following Eq. (2). A rainflow counter was used to determine the number of cycles ($n$) for different amplitude oscillations of each output QOI. A constant Wöhler exponent ($w$) was used. Output QOI pertaining to composite components used a value of $w = 10.0$, and QOI pertaining to steel elements used a value of $w = 4.0$. $N$ is the number of cycles for the equivalent load, and a value of 3600 was used for a 1 Hz DEL given the 3600 s non-transient time. The value $N$ scales all loads equally and has no effect on the final fatigue load sensitivity conclusions.

$$Y_{fat.}(U) = \frac{1}{S} \sum_{s=1}^{S} (\sum \frac{n}{N} Y(U)^w)^{\frac{1}{w}} \tag{2}$$

Ultimate EE values were calculated following Eq. (3), where $U^b$ represents the set of input parameters at some starting point, $b$, and $x_i$ is a perturbation in only one input parameter, $i$. The EE value is normalized with the total range of the input parameter





for that load case, so that the sensitivity can be compared between different input parameters. For the fixed perturbation size in this project of 10%, the change in output QOI is effectively multiplied by 10. Ultimate EE values include the addition of the QOI value based on all nominal values for a given load case, $\overline{Y_{LC}}$. This allows the ultimate load sensitivities to be compared across multiple load cases. If a specific load case does not lead to critical ultimate loads for a specific QOI, the sensitivity is less likely to be relatively important.

$$UEE_{iLC}^{b} = \left| \frac{Y_{ult.}(U^b + x_i) - Y_{ult.}(U^b)}{\Delta_{iLC}} u_{iLC,range} \right| + \overline{Y_{LC}} \tag{3}$$

Fatigue EE values were calculated in a similar way, following Eq. (4). No nominal value was added to the fatigue value, as the focus is on the variability of the loading. There is also only one load case in this study that considers fatigue loads.

$$FEE_{iLC}^{b} = \left| \frac{Y_{fat.}(U^b + x_i) - Y_{fat.}(U^b)}{\Delta_{iLC}} u_{iLC,range} \right| \tag{4}$$

For a given QOI, all relevant load case EE results were compared. Significant EE values were defined as those larger than a threshold of two standard deviations above the mean. These significant EE values were counted, and the relative number of significant EE values coming from perturbations in a certain input parameter indicate the sensitivity to that input.

## 3 Input Parameters

A set of 40 input parameters were selected based on expectations of possible sensitivities. These include parameters that were found to be important in previous studies and additions that were expected to be potentially relevant to above-cut-out conditions. The list of selected variables is shown in Table 2, and the corresponding labels are used in the following text and figures.





**Table 2.** Variable input parameters and labels for following text and figures

| Wind | | | Turbine | | | Water | | |
|---|---|---|---|---|---|---|---|---|
| | Mean wind speed | $U_{mean}$ | | Yaw Error | $Yaw$ | | Depth | $Depth$ |
| | Shear exponent | $Shear$ | | Blade twist at root | $Twist_{root}$ | | Directional spreading | $\Theta_{spread}$ |
| | Veer | $Veer$ | | Blade twist at tip | $Twist_{tip}$ | | Wind-wave misalignment | $\Theta_{mis}$ |
| | Coherence exponent | $\gamma_{wind}$ | | | | | Significant wave height | $Hs$ |
| | Standard deviation | $\sigma_u$ | Structure | Tower stiffness | $TK$ | | Maximum wave height | $H_{max}$ |
| | Integral-scale parameter | $L_u$ | | Tower damping | $TD$ | | Position of maximum wave height | $XH_{max}$ |
| | Coherence decrement | $a_u$ | | System COG X | $COG_X$ | | Peak wave period | $Tp$ |
| | Offset parameter | $b_u$ | | System COG Z | $COG_Z$ | | Wave spectral shape factor | $\gamma_{wave}$ |
| | | | | System pitch/roll inertia | $IYY$ | | Current speed | $V_{current}$ |
| Aerodynamic | Lift coefficient at blade tip | $Cl_{tip}$ | | System yaw inertia | $IZZ$ | | Current direction | $\Theta_{current}$ |
| | Drag coefficient at blade tip | $Cd_{tip}$ | | System mass | $Mass$ | | | |
| | Lift coefficient at blade root | $Cl_{root}$ | | | | **Hydrodynamic** | Drag coefficient of upper column | $Cd_{upper}$ |
| | Drag coefficient at blade root | $Cd_{root}$ | Mooring | Polyester axial stiffness | $K_{fiber}$ | | Drag coefficient of lower column | $Cd_{lower}$ |
| | Drag coefficient of tower | $Cd_{tower}$ | | Polyester length | $L_{fiber}$ | | Axial drag coefficient of column | $Cd_{axial}$ |
| | | | | Chain linear density | $M_{chain}$ | | Drag coefficient of rectangular base | $Cd_{rectangular}$ |

Definitions of the wind parameters are the same as the 2018 study that focused fully on the incident wind (Robertson et al., 2018).

Aerodynamic lift and drag coefficients for the blade were dictated at the tip and root. The range denotes a fraction change from the defined value for the airfoil. The coefficients at the mid-span sections between the tip and the root use a coefficient modification that is linearly interpolated between the tip and root. The range given for $Cd_{tower}$ is the absolute nondimensional coefficient.

$Yaw$ is the misalignment angle between the rotor nacelle assembly and the incident wind at hub height. Blade twist is altered both at the tip and the root. $Twist_{root}$ is effectively an error in the blade pitch, and the relative difference between $Twist_{tip}$ and $Twist_{root}$ is some error in geometric twist.

$TK$ is a multiplier of the tower stiffness in both the fore-aft and side-to-side directions. $TD$ is a percentage of the critical damping ratio. The system mass and inertia properties are varied collectively for the platform and the nacelle. Mooring parameters impact the polyester and chain sections individually.

Definitions of the water parameters are the same as the 2023 study that focused on a floating wind system in operational conditions (Wiley et al., 2023). $H_{max}$ and $XH_{max}$ are new additions to this study and are defined in Sect. 3.1.3.

Platform viscous drag coefficients were varied separately for different components of the floater. There are separate variable inputs for the columns near the waterline, $Cd_{upper}$, the columns in deeper water, $Cd_{lower}$, the ends of the columns, $Cd_{axial}$, and the rectangular horizontal members, $Cd_{rectangular}$.





## 3.1 Parameter Ranges

The range of each parameter was selected for each load case and is shown in Table 3 for the Gulf of Maine and in Table 4 for Humboldt Bay. Ultimate load cases include nominal values to use for combining load cases, as described in Eq. (3). Descriptions of the justification in parameter ranges is given in the following subsections of Sect. 3.1.





**Table 3.** Modeling input parameter ranges and nominal values for the Gulf of Maine

| | | | 6.1 | | | 6.3 | | | 6.4 | | 6.5 | | |
|---|---|---|---|---|---|---|---|---|---|---|---|---|---|
| | | | Min. | Nom. | Max. | Min. | Nom. | Max. | Min. | Max. | Min. | Nom. | Max. |
| $U_{mean}$ | | m s$^{-1}$ | 37.58 | 38.96 | 40.04 | 33.84 | 34.57 | 35.45 | 25.00 | 28.03 | 37.91 | 40.32 | 43.42 |
| $Shear$ | | - | 0.082 | 0.110 | 0.167 | 0.082 | 0.110 | 0.185 | 0.057 | 0.221 | 0.082 | 0.110 | 0.163 |
| $Veer$ | | deg m$^{-1}$ | 0.0000 | 0.0000 | 0.0249 | 0.0000 | 0.0000 | 0.0332 | 0.0000 | 0.0616 | 0.0000 | 0.0000 | 0.0249 |
| $\gamma_{wind}$ | | - | 0.0 | 0.0 | 1.0 | 0.0 | 0.0 | 1.0 | 0.0 | 1.0 | 0.0 | 0.0 | 1.0 |
| $\sigma_u$ | | m s$^{-1}$ | 3.546 | 3.701 | 4.013 | 3.146 | 3.284 | 3.561 | 2.413 | 2.731 | 3.669 | 3.830 | 4.153 |
| $L_u$ | | m | 25.0 | 340.2 | 1600.0 | 25.0 | 340.2 | 1600.0 | 25.0 | 1600.0 | 25.0 | 340.2 | 1600.0 |
| $a_u$ | | - | 1.5 | 12 | 26 | 1.5 | 12 | 26 | 1.5 | 26 | 1.5 | 12 | 26 |
| $b_u$ | | m$^{-1}$ | 0 | 0.000353 | 0.05 | 0 | 0.000353 | 0.05 | 0 | 0.05 | 0 | 0.000353 | 0.05 |
| $Cl_{tip}$ | | - | -0.26 | 0.0 | 0.26 | -0.26 | 0.0 | 0.26 | -0.26 | 0.26 | -0.26 | 0.0 | 0.26 |
| $Cd_{tip}$ | | - | -1.0 | 0.0 | 1.0 | -1.0 | 0.0 | 1.0 | -1.0 | 1.0 | -1.0 | 0.0 | 1.0 |
| $Cl_{root}$ | | - | -0.26 | 0.0 | 0.26 | -0.26 | 0.0 | 0.26 | -0.26 | 0.26 | -0.26 | 0.0 | 0.26 |
| $Cd_{root}$ | | - | -1.0 | 0.0 | 1.0 | -1.0 | 0.0 | 1.0 | -1.0 | 1.0 | -1.0 | 0.0 | 1.0 |
| $Cd_{nacelle}$ | | - | 1 | 1.2 | 1.4 | 1 | 1.2 | 1.4 | 1 | 1.4 | 1 | 1.2 | 1.4 |
| $Yaw$ | | ° | -8 | 0 | 8 | -20 | 0 | 20 | -8 | 8 | -8 | 0 | 8 |
| $Twist_{root}$ | | ° | -2.0 | 0.0 | 2.0 | -2.0 | 0.0 | 2.0 | -2.0 | 2.0 | -2.0 | 0.0 | 2.0 |
| $Twist_{tip}$ | | ° | -2.0 | 0.0 | 2.0 | -2.0 | 0.0 | 2.0 | -2.0 | 2.0 | -2.0 | 0.0 | 2.0 |
| $TK$ | | - | 0.707 | 1 | 1.216 | 0.707 | 1 | 1.216 | 0.707 | 1.216 | 0.707 | 1 | 1.216 |
| $TD$ | | - | 0.1 | 1 | 5 | 0.1 | 1 | 5 | 0.1 | 5 | 0.1 | 1 | 5 |
| $COG_X$ | Ptfm | m | -1.035 | 0.0 | 1.035 | -1.035 | 0.0 | 1.035 | -1.035 | 1.035 | -1.035 | 0 | 1.035 |
| | Nac | m | -4.8145 | -4.7201 | -4.6257 | -4.8145 | -4.7201 | -4.6257 | -4.8145 | -4.6257 | -4.8145 | -4.7201 | -4.6257 |
| $COG_Z$ | Ptfm | m | -15.1 | -14.4 | -13.7 | -15.1 | -14.4 | -13.7 | -15.1 | -13.7 | -15.1 | -14.4 | -13.7 |
| | Nac | m | 4.1896 | 4.2751 | 4.3606 | 4.1896 | 4.2751 | 4.3606 | 4.1896 | 4.3606 | 4.1896 | 4.2751 | 4.3606 |
| $IYY$ | Ptfm | kg m$^{-2}$ | 1.23E+10 | 1.25E+10 | 1.28E+10 | 1.23E+10 | 1.25E+10 | 1.28E+10 | 1.23E+10 | 1.28E+10 | 1.23E+10 | 1.25E+10 | 1.28E+10 |
| $IZZ$ | Ptfm | kg m$^{-2}$ | 2.32E+10 | 2.37E+10 | 2.41E+10 | 2.32E+10 | 2.37E+10 | 2.41E+10 | 2.32E+10 | 2.41E+10 | 2.32E+10 | 2.37E+10 | 2.41E+10 |
| | Nac | kg m$^{-2}$ | 2.38E+07 | 2.42E+07 | 2.47E+07 | 2.38E+07 | 2.42E+07 | 2.47E+07 | 2.38E+07 | 2.47E+07 | 2.38E+07 | 2.42E+07 | 2.47E+07 |
| $Mass$ | Ptfm | kg | 1.75E+07 | 1.78E+07 | 1.82E+07 | 1.75E+07 | 1.78E+07 | 1.82E+07 | 1.75E+07 | 1.82E+07 | 1.75E+07 | 1.78E+07 | 1.82E+07 |
| | Nac | kg | 6.34E+05 | 6.47E+05 | 6.60E+05 | 6.34E+05 | 6.47E+05 | 6.60E+05 | 6.34E+05 | 6.60E+05 | 6.34E+05 | 6.47E+05 | 6.60E+05 |
| $K_{fiber}$ | Stat | N m$^{-1}$ | 1.39E+08 | 1.42E+08 | 1.46E+08 | 1.39E+08 | 1.42E+08 | 1.46E+08 | 1.39E+08 | 1.46E+08 | 1.39E+08 | 1.42E+08 | 1.46E+08 |
| | Dyn | N m$^{-1}$ | 1.55E+08 | 1.59E+08 | 1.63E+08 | 1.55E+08 | 1.59E+08 | 1.63E+08 | 1.55E+08 | 1.63E+08 | 1.55E+08 | 1.59E+08 | 1.63E+08 |
| $L_{fiber}$ | | m | 194.81 | 199.80 | 204.80 | 194.81 | 199.80 | 204.80 | 194.81 | 204.80 | 194.81 | 199.80 | 204.80 |
| $M_{chain}$ | | kg m$^{-1}$ | 468.91 | 480.93 | 492.95 | 468.91 | 480.93 | 492.95 | 468.91 | 492.95 | 468.91 | 480.93 | 492.95 |
| $Depth$ | | m | 197.6 | 200.0 | 202.7 | 197.6 | 200.0 | 202.7 | 198.4 | 201.6 | 197.6 | 200.0 | 202.7 |
| $\Theta_{spread}$ | | ° | 0 | 0 | 20 | 0 | 0 | 20 | 0 | 20 | 0 | 0 | 20 |
| $Theta_{mis}$ | | ° | 0.0 | 0.0 | 152.4 | 0.0 | 0.0 | 161.3 | 0.0 | 173.9 | 0.0 | 0.0 | 150.5 |
| $Hs$ | | m | 9.27 | 10.56 | 11.46 | 6.90 | 7.13 | 7.39 | 1.88 | 6.55 | 9.98 | 12.47 | 14.72 |
| $H_{max}$ | | - | 1.75 | 1.86 | 2.15 | 1.75 | 1.86 | 2.15 | 1.75 | 2.15 | 1.75 | 1.86 | 2.15 |
| $XH_{max}$ | | - | -λ/2 | 0 | λ/2 | -λ/2 | 0 | λ/2 | -λ/2 | λ/2 | -λ/2 | 0 | λ/2 |
| $Tp$ | | s | f(Hs) | f(Hs) | f(Hs) | f(Hs) | f(Hs) | f(Hs) | f(Hs) | f(Hs) | f(Hs) | f(Hs) | f(Hs) |
| $\gamma_{wave}$ | | - | 1 | f(Hs,Tp) | 7 | 1 | f(Hs,Tp) | 7 | 1 | 7 | 1 | f(Hs,Tp) | 7 |
| $V_{current}$ | | m s$^{-1}$ | 0.87 | 1.20 | 1.75 | 0.58 | 0.63 | 0.66 | 0.08 | 0.71 | 0.96 | 1.64 | 3.00 |
| $\Theta_{current}$ | | ° | 0.0 | 0.0 | 180.0 | 0.0 | 0.0 | 180.0 | 0.0 | 180.0 | 0.0 | 0.0 | 180.0 |
| $Cd_{upper}$ | | - | 0.4 | 1.3 | 2.0 | 0.4 | 1.3 | 2.0 | 0.4 | 2.0 | 0.4 | 1.3 | 2.0 |
| $Cd_{lower}$ | | - | 0.4 | 0.4 | 2.0 | 0.4 | 0.4 | 2.0 | 0.4 | 2.0 | 0.4 | 0.4 | 2.0 |
| $Cd_{axial}$ | | - | 0.7 | 8.0 | 10.0 | 0.7 | 8.0 | 10.0 | 0.7 | 10.0 | 0.7 | 8.0 | 10.0 |
| $Cd_{rectangular}$ | | - | 0.7 | 4.0 | 10.0 | 0.7 | 4.0 | 10.0 | 0.7 | 10.0 | 0.7 | 4.0 | 10.0 |



**Table 4.** Modeling input parameter ranges and nominal values for Humboldt Bay

| | | | 6.1 | | | 6.3 | | | 6.4 | | 6.5 | | |
|---|---|---|---|---|---|---|---|---|---|---|---|---|---|
| | | | Min. | Nom. | Max. | Min. | Nom. | Max. | Min. | Max. | Min. | Nom. | Max. |
| $U_{mean}$ | | m s$^{-1}$ | 33.97 | 38.33 | 41.03 | 31.35 | 31.95 | 33.20 | 25.00 | 28.72 | 34.29 | 40.59 | 49.90 |
| $Shear$ | | - | 0.108 | 0.110 | 0.148 | 0.111 | 0.110 | 0.197 | 0.100 | 0.184 | 0.109 | 0.110 | 0.121 |
| $Veer$ | | deg m$^{-1}$ | 0.0000 | 0.0000 | 0.0048 | 0.0000 | 0.0000 | 0.0139 | -0.0033 | 0.0222 | -0.0003 | 0.0000 | 0.0039 |
| $\gamma_{wind}$ | | - | 0.0 | 0.0 | 1.0 | 0.0 | 0.0 | 1.0 | 0.0 | 1.0 | 0.0 | 0.0 | 1.0 |
| $\sigma_u$ | | m s$^{-1}$ | 3.066 | 3.641 | 4.676 | 2.556 | 3.035 | 3.898 | 2.149 | 3.277 | 3.247 | 3.856 | 4.952 |
| $L_u$ | | m | 25.000 | 340.200 | 1600.000 | 25.0 | 340.2 | 1600.0 | 25.0 | 1600.0 | 25.0 | 340.2 | 1600.0 |
| $a_u$ | | - | 1.5 | 12 | 26.0 | 1.5 | 12 | 26 | 1.5 | 26 | 1.5 | 12 | 26 |
| $b_u$ | | m$^{-1}$ | 0 | 0.000353 | 0.05 | 0 | 0.000353 | 0.05 | 0 | 0.05 | 0 | 0.000353 | 0.05 |
| $Cl_{tip}$ | | - | -0.26 | 0.0 | 0.26 | -0.26 | 0.0 | 0.26 | -0.26 | 0.26 | -0.26 | 0.0 | 0.26 |
| $Cd_{tip}$ | | - | -1.0 | 0.0 | 1.0 | -1.0 | 0.0 | 1.0 | -1.0 | 1.0 | -1.0 | 0.0 | 1.0 |
| $Cl_{root}$ | | - | -0.26 | 0.0 | 0.26 | -0.26 | 0.0 | 0.26 | -0.26 | 0.26 | -0.26 | 0.0 | 0.26 |
| $Cd_{root}$ | | - | -1.0 | 0.0 | 1.0 | -1.0 | 0.0 | 1.0 | -1.0 | 1.0 | -1.0 | 0.0 | 1.0 |
| $Cd_{nacelle}$ | | - | 1 | 1.2 | 1.4 | 1 | 1.2 | 1.4 | 1 | 1.4 | 1 | 1.2 | 1.4 |
| $Yaw$ | | ° | -8 | 0 | 8 | -20 | 0 | 20 | -8 | 8 | -8 | 0 | 8 |
| $Twist_{root}$ | | ° | -2.0 | 0.0 | 2.0 | -2.0 | 0.0 | 2.0 | -2.0 | 2.0 | -2.0 | 0.0 | 2.0 |
| $Twist_{tip}$ | | ° | -2.0 | 0.0 | 2.0 | -2.0 | 0.0 | 2.0 | -2.0 | 2.0 | -2.0 | 0.0 | 2.0 |
| $TK$ | | - | 0.707 | 1 | 1.216 | 0.707 | 1 | 1.216 | 0.707 | 1.216 | 0.707 | 1 | 1.216 |
| $TD$ | | - | 0.1 | 1 | 5 | 0.1 | 1 | 5 | 0.1 | 5 | 0.1 | 1 | 5 |
| $COG_X$ | Ptfm | m | -1.035 | 0.0 | 1.035 | -1.035 | 0.0 | 1.035 | -1.035 | 1.035 | -1.035 | 0 | 1.035 |
| | Nac | m | -4.8145 | -4.7201 | -4.6257 | -4.8145 | -4.7201 | -4.6257 | -4.8145 | -4.6257 | -4.8145 | -4.7201 | -4.6257 |
| $COG_Z$ | Ptfm | m | -15.1 | -14.4 | -13.7 | -15.1 | -14.4 | -13.7 | -15.1 | -13.7 | -15.1 | -14.4 | -13.7 |
| | Nac | m | 4.1896 | 4.2751 | 4.3606 | 4.1896 | 4.2751 | 4.3606 | 4.1896 | 4.3606 | 4.1896 | 4.2751 | 4.3606 |
| $IYY$ | Ptfm | kg m$^{-2}$ | 1.23E+10 | 1.25E+10 | 1.28E+10 | 1.23E+10 | 1.25E+10 | 1.28E+10 | 1.23E+10 | 1.28E+10 | 1.23E+10 | 1.25E+10 | 1.28E+10 |
| $IZZ$ | Ptfm | kg m$^{-2}$ | 2.32E+10 | 2.37E+10 | 2.41E+10 | 2.32E+10 | 2.37E+10 | 2.41E+10 | 2.32E+10 | 2.41E+10 | 2.32E+10 | 2.37E+10 | 2.41E+10 |
| | Nac | kg m$^{-2}$ | 23756096 | 24240914 | 24725732 | 2.38E+07 | 2.42E+07 | 2.47E+07 | 2.38E+07 | 2.47E+07 | 2.38E+07 | 2.42E+07 | 2.47E+07 |
| $Mass$ | Ptfm | kg | 17481240 | 17838000 | 18194760 | 1.75E+07 | 1.78E+07 | 1.82E+07 | 1.75E+07 | 1.82E+07 | 1.75E+07 | 1.78E+07 | 1.82E+07 |
| | Nac | kg | 633957.1 | 646895 | 659832.9 | 6.34E+05 | 6.47E+05 | 6.60E+05 | 6.34E+05 | 6.60E+05 | 6.34E+05 | 6.47E+05 | 6.60E+05 |
| $K_{fiber}$ | Stat | N m$^{-1}$ | 1.42E+08 | 1.46E+08 | 1.49E+08 | 1.42E+08 | 1.46E+08 | 1.49E+08 | 1.42E+08 | 1.49E+08 | 1.42E+08 | 1.46E+08 | 1.49E+08 |
| | Dyn | N m$^{-1}$ | 1.99E+08 | 2.04E+08 | 2.09E+08 | 1.99E+08 | 2.04E+08 | 2.09E+08 | 1.99E+08 | 2.09E+08 | 1.99E+08 | 2.04E+08 | 2.09E+08 |
| $L_{fiber}$ | | m | 1344.45 | 1378.92 | 1413.39 | 1344.45 | 1378.92 | 1413.39 | 1344.45 | 1413.39 | 1344.45 | 1378.92 | 1413.39 |
| $M_{chain}$ | | kg m$^{-1}$ | 280.80 | 288.00 | 295.20 | 280.80 | 288.00 | 295.20 | 280.80 | 295.20 | 280.80 | 288.00 | 295.20 |
| $Depth$ | | m | 797.8 | 800.0 | 802.4 | 797.8 | 800.0 | 802.4 | 798.5 | 801.5 | 797.8 | 800.0 | 802.4 |
| $\Theta_{spread}$ | | ° | 0 | 0 | 20 | 0 | 0 | 20 | 0 | 20 | 0 | 0 | 20 |
| $Theta_{mis}$ | | ° | 0.0 | 0.0 | 171.0 | 0.0 | 0.0 | 177.6 | 0.0 | 168.4 | 0.0 | 0.0 | 170.1 |
| $Hs$ | | m | 10.68 | 11.48 | 11.97 | 8.27 | 8.54 | 8.87 | 3.05 | 6.07 | 11.07 | 12.44 | 13.35 |
| $H_{max}$ | | - | 1.75 | 1.86 | 2.15 | 1.75 | 1.86 | 2.15 | 1.75 | 2.15 | 1.75 | 1.86 | 2.15 |
| $XH_{max}$ | | - | $-\lambda/2$ | 0 | $\lambda/2$ | $-\lambda/2$ | 0 | $\lambda/2$ | $-\lambda/2$ | $\lambda/2$ | $-\lambda/2$ | 0 | $\lambda/2$ |
| $Tp$ | | s | f(Hs) | f(Hs) | f(Hs) | f(Hs) | f(Hs) | f(Hs) | f(Hs) | f(Hs) | f(Hs) | f(Hs) | f(Hs) |
| $\gamma_{wave}$ | | - | 1 | f(Hs,Tp) | 7 | 1 | f(Hs,Tp) | 7 | 1 | 7 | 1 | f(Hs,Tp) | 7 |
| $V_{current}$ | | m s$^{-1}$ | 0.99 | 1.00 | 1.00 | 0.98 | 0.99 | 0.99 | 0.03 | 0.47 | 0.99 | 1.00 | 1.01 |
| $\Theta_{current}$ | | ° | 0.0 | 0.0 | 180.0 | 0.0 | 0.0 | 180.0 | 0.0 | 180.0 | 0.0 | 0.0 | 180.0 |
| $Cd_{upper}$ | | - | 0.4 | 1.3 | 2.0 | 0.4 | 1.3 | 2.0 | 0.4 | 2.0 | 0.4 | 1.3 | 2.0 |
| $Cd_{lower}$ | | - | 0.4 | 0.4 | 2.0 | 0.4 | 0.4 | 2.0 | 0.4 | 2.0 | 0.4 | 0.4 | 2.0 |
| $Cd_{axial}$ | | - | 0.7 | 8.0 | 10.0 | 0.7 | 8.0 | 10.0 | 0.7 | 10.0 | 0.7 | 8.0 | 10.0 |
| $Cd_{rectangular}$ | | - | 0.7 | 4.0 | 10.0 | 0.7 | 4.0 | 10.0 | 0.7 | 10.0 | 0.7 | 4.0 | 10.0 |



### 3.1.1 Wind and Aerodynamic

Environmental conditions are defined in the IEC load case definitions; ultimate load cases use extreme values with a specified return period. There is uncertainty in these extreme values for a number of reasons, including extrapolation of data from measurement location to deployment location, measurement errors, and a limited length of measurement history. The length of the dataset is expected to be particularly important for the calculation of extreme values. The extreme environmental condition ranges in Tables 3 and 4 are a quantification of this uncertainty.

Wind data for both locations came from the 2023 National Offshore Wind Data Set (NOW23) (Bodini et al., 2023). The dataset was generated for eight U.S. offshore regions using the Weather Research and Forecasting model, which utilizes data from offshore lidars, buoys, and coastal radars (Bodini et al., 2023). In the Gulf of Maine, 21 years of data are available, and in Humboldt Bay, 23 years of data are available.

Extreme values of the wind parameters at these locations were calculated with the generalized extreme value (GEV) distribution using a block maxima approach with a block size of 1 year. Using a block maxima reduces the number of data points to fit the distribution but takes the seasonality of the environment into consideration. The probability of an extreme event should not change from one block to another, introducing error if smaller block sizes are used; a peaks-over-threshold approach has a similar challenge. There is some uncertainty in the fit and selection of a distribution. For a given distribution, a bootstrapping approach is commonly used to quantify a confidence interval of an extreme value estimation (Vanem, 2015). This is done by recalculating the distribution with a random sample with replacement from the dataset many times. Each time the distribution is calculated, the extreme values are calculated from the sample, and a distribution of extreme values is formed. If the data are very consistent, or if the number of data points is very large, the distribution of extreme values is narrow. If the data are not consistent, or the number of data points is small, the distribution of extreme values is wide. This approach was used for determining the range of uncertainty in the extreme values of $U_{mean}$. 1500 resamples were performed of the GEV distribution. The distributions of extreme values with the relevant return periods are shown in Fig. 4. The dashed red line marks the nominal value, which is calculated using the full dataset with no resampling. The two solid lines show the 0.1 and 0.9 quantiles, which were selected as the limits of the input parameter range for $U_{mean}$. The range of uncertainty grows with the return period and is generally larger for Humboldt Bay than for the Gulf of Maine.



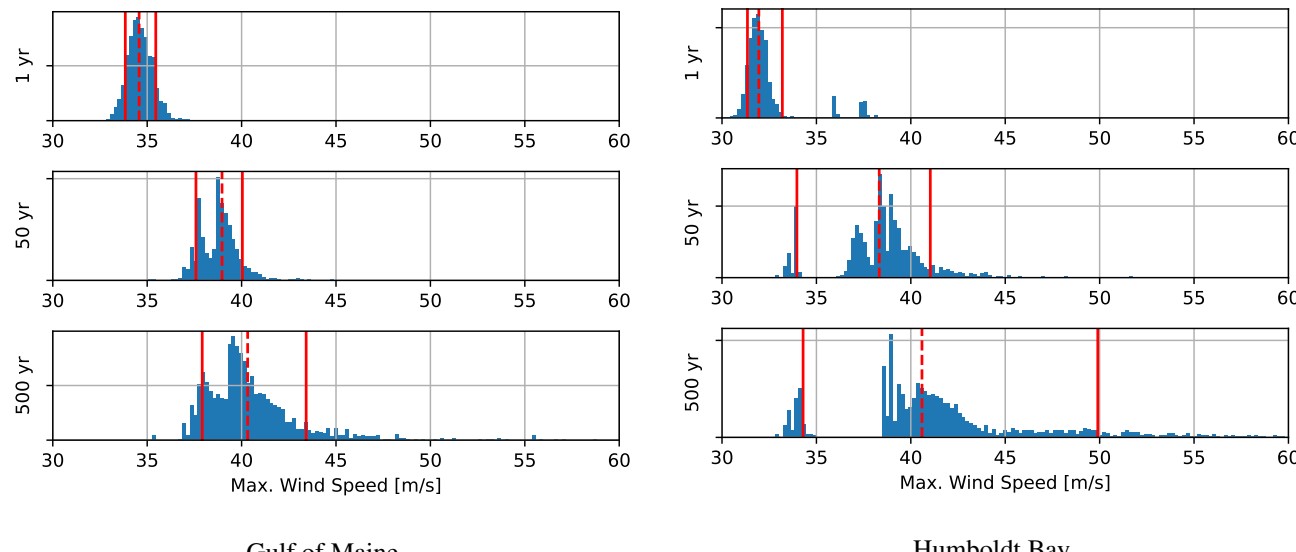

Gulf of Maine            Humboldt Bay

**Figure 4.** Bootstrapped distribution of $U_{mean}$ with return periods of 1, 50, and 500 years based on a generalized extreme value distribution for the Gulf of Maine (left) and Humboldt Bay (right). The nominal value is based on the full dataset with no resampling and is shown with a dashed red line. The limits of the parameter range are the 0.1 and 0.9 quantiles of the bootstrapped distribution and are shown with solid red lines.

The $Shear$ was also extracted from the NOW23 dataset. The vertical wind speed distribution was fit to a power law function, and the exponent was extracted at each time step. It is expected that the amount of $Shear$ in the flow is a function of the $U_{mean}$. The ranges were selected based on the time steps with a value of $U_{mean}$ within the range for a given load case. Figure 5 shows a normalized histogram of $Shear$ over all recorded time and normalized histograms for the time corresponding to each load case.





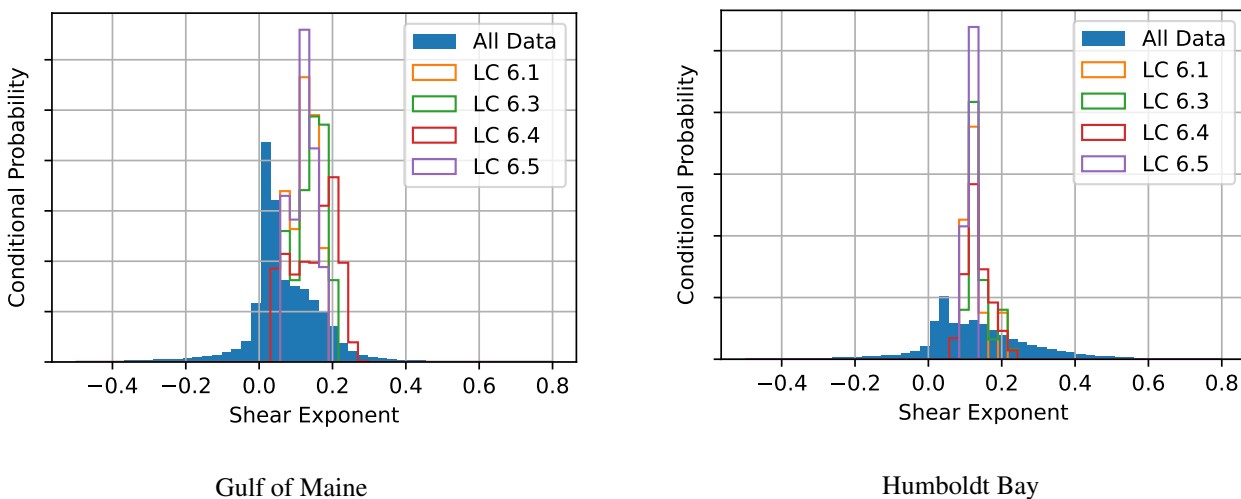

Gulf of Maine          Humboldt Bay

**Figure 5.** $Shear$ with conditional probability based on $U_{mean}$ ranges corresponding to IEC design load cases 6.1, 6.3, 6.4, and 6.5 for the Gulf of Maine (left) and Humboldt Bay (right)

The $Veer$ was extracted from the NOW23 dataset assuming a linear fit over the elevations of the undisplaced rotor. The range for $Veer$ was also chosen based on the time with a value of $U_{mean}$ within the corresponding range for each load case. Figure 6 shows normalized histograms for the full dataset and each load case.

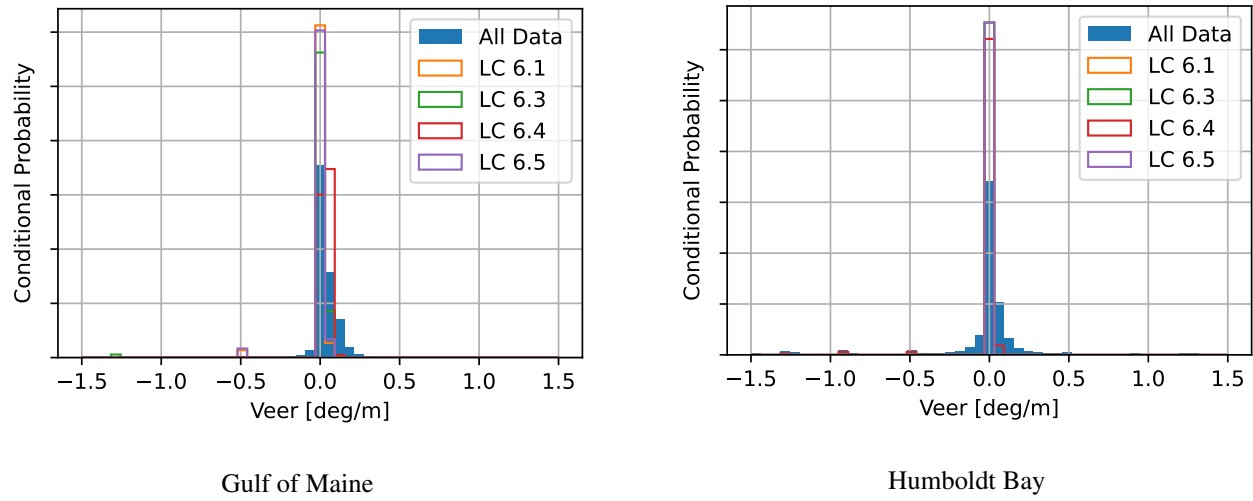

Gulf of Maine          Humboldt Bay

**Figure 6.** $Veer$ with conditional probability based on $U_{mean}$ ranges corresponding to IEC design load cases 6.1, 6.3, 6.4, and 6.5 for the Gulf of Maine (left) and Humboldt Bay (right)

     The turbulence intensity ranges, defined by the value of $\sigma_u$, are based on published local measured ranges. The data for the Gulf of Maine came from a combination of land-based and offshore lidar deployed by the University of Maine (Viselli et al.,

2018). The data for Humboldt Bay came from a buoy deployed by the Pacific Northwest National Laboratory (Krishnamurthy





et al., 2023). The values from both sites are published as a function of $U_{mean}$; however, neither function goes to high enough values of $U_{mean}$ to cover the load cases of this work. At both locations, the value of turbulence intensity starts to converge with increasing $U_{mean}$, and the ranges were taken from the maximum provided speed. The turbulence intensity range was applied to the nominal value of $U_{mean}$ for each load case to determine the $\sigma_u$ range for the load case.

No site-specific information was available to describe $L_u$, $a_u$, or $b_u$. Ranges matching those of the highest $U_{mean}$ load cases from previous EE studies were used (Robertson et al., 2018) (Robertson et al., 2019) (Shaler et al., 2021) (Wiley et al., 2023).

     The range for the aerodynamic coefficients is meant to be an aggregate of uncertainties in the steady polars and is consistent with previous EE studies focusing on operational load cases (Shaler et al., 2019).

### 3.1.2    System and Structure

The range for $Yaw$, the angle between the wind direction and the turbine orientation, is dictated by the IEC load cases for each of the ultimate load cases for LC 6.1 and 6.3. LC 6.3 is specifically configured to check the loads with extreme $Yaw$; the same smaller range used in LC 6.1 is also applied to LC 6.4 and 6.5 (IEC, 2024).

     The ranges for $Twist_{root}$ and $Twist_{tip}$ capture collective blade pitch error and the sensitivity to some error in twist, either due to manufacturing uncertainty, changes with time, or non-captured torsional displacement (Petrone et al., 2011).

The tower stiffness was selected to result in a $\pm15\%$ change in tower first modal frequency. The change to the stiffness was verified with a simplified clamped boundary condition test using BModes. Ranges for system mass and inertia were $\pm2\%$ of the design values. This range of uncertainty was suggested by industry experts and was applied in a previous operational EE study (Wiley et al., 2023). Mooring line ranges for the fiber sections represent an aggregate of uncertainty in the original properties and changes throughout a project lifetime due to creep. The range for $K_{fiber}$ is $\pm2.5\%$ of the design values, and 245 the static and dynamic stiffnesses are changed together. The ranges for $L_{fiber}$ and $M_{chain}$ are also $\pm2.5\%$. These ranges were suggested by mooring industry experts, and the changes in fiber properties through time, in particular, were highlighted as an area that needs future work.

### 3.1.3    Water and Hydrodynamic

The ranges for $Depth$ are based on changes in time due to the water level, and the water level range is dictated by the IEC 250 as either normal for the fatigue load cases or extreme for the ultimate load cases (IEC, 2024). Local bathymetry changes in a mooring footprint are expected to be accounted for in design and installation, while changes due to water level variation will occur in all cases. The water level range in the Gulf of Maine was described in a characterization of the site (Krieger et al., 2015). The range for Humboldt Bay was extrapolated from the ArcGIS tidal range map (ArcGIS, 2024).

     Wave data came from the National Oceanic and Atmospheric Administration's National Data Buoy Center (NDBC). Data 255 covering 45 years from buoy 44005 were used for the Gulf of Maine, and data covering 41 years from buoy 46022 were used for Humboldt Bay (NOAA, 2024).

     No site-specific data were found to quantify the range of directional spreading, $\Theta_{spread}$, so a maximum range of $20°$ was assumed. The angle between the wind direction, from the NOW23 dataset, and the direction of the wave propagation, from the





NDBC, were available to quantify $\Theta_{mis}$. Similar to the wind $Shear$ and $Veer$, the values of $\Theta_{mis}$ were grouped by load case

based on the concurrent value of $U_{mean}$. The resulting normalized histograms are shown in Fig. 7.

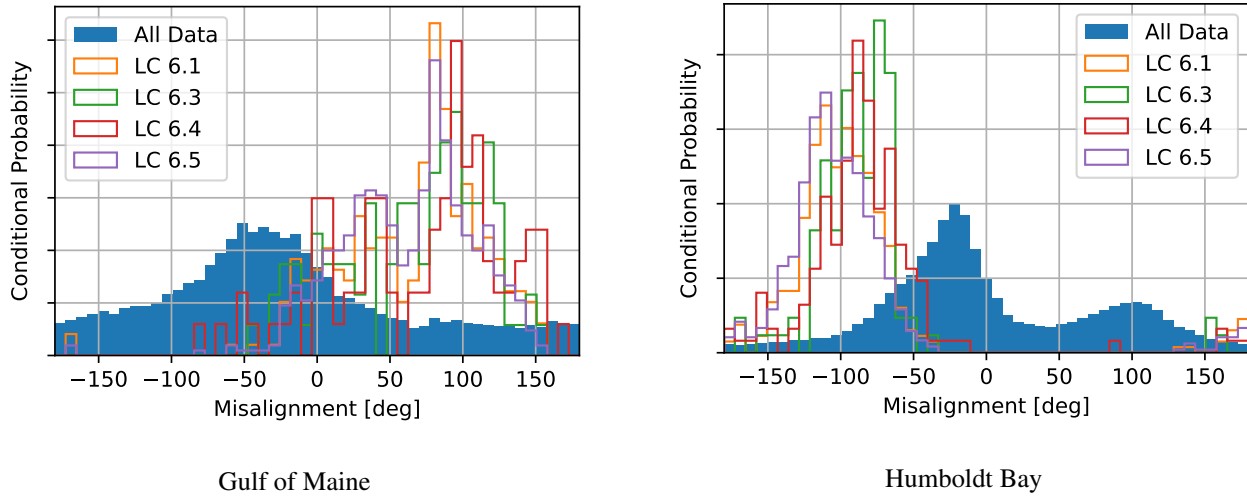

**Figure 7.** $\theta_{mis}$ with conditional probability based on wind speed ranges corresponding to IEC design load cases 6.1, 6.3, 6.4, and 6.5 for the Gulf of Maine (left) and Humboldt Bay (right)

The range of uncertainty for extreme $Hs$ was determined with the same GEV bootstrapped approach as the extreme $U_{mean}$. The resulting distributions of extreme values and the quantile ranges are shown in Fig. 8. The distribution at Humboldt Bay is narrower, resulting in a smaller range. There is a bimodal nature to the data from Humboldt Bay; this is much more apparent with larger return periods.



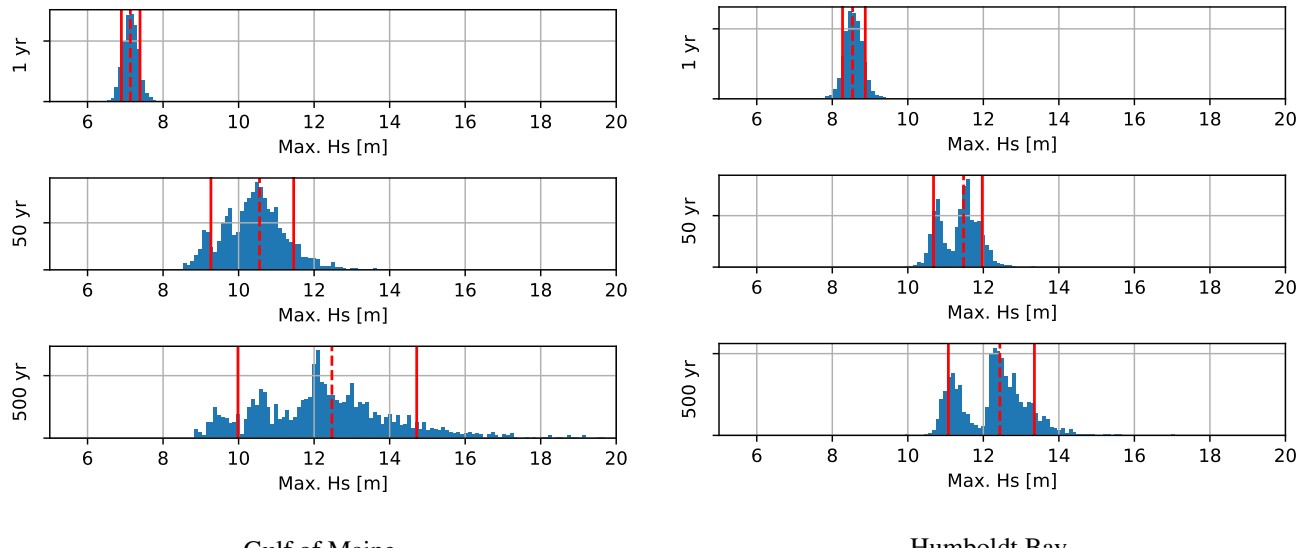

Gulf of Maine          Humboldt Bay

**Figure 8.** Bootstrapped distribution of $Hs$ with return periods of 1, 50, and 500 years based on a generalized extreme value distribution for the Gulf of Maine (left) and Humboldt Bay (right). The nominal value is based on the full dataset with no resampling and is shown with a dashed red line. The limits of the parameter range are the 0.1 and 0.9 quantiles of the bootstrapped distribution and are shown with solid red lines.

An embedded constrained wave of a defined maximum height is available in the SeaState module of OpenFAST. It guarantees that some maximum wave height will occur in the modeled, otherwise irregular, time series, allowing a check of the system robustness to an extreme wave with a shorter physical time. The height, location, and the time of the embedded wave can be specified. In this study, the time was held constant, but the height and location were varied. The nominal value of $H_{max}$ comes from the IEC recommendation and is based on a value of 0.1% exceedance with 1000 waves and a Rayleigh distribution (IEC,

2024). $H_{max}$ is a multiplier of $Hs$. The limits chosen use the same 0.1% exceedance according to a Rayleigh distribution, but for a 3 h sea state with a range of average periods corresponding to expected wave periods. The position of the maximum wave crest, $XH_{max}$, is a function of the $Tp$. All possible phases of the wave were tested with a range of $\pm\frac{1}{2}$ of the wavelength.

    The wave $Tp$ was treated as a function of $Hs$ for each combination of input parameters. The processed data from NDBC have discrete values of $Tp$ based on the frequency resolution used in the wave spectra. Gaussian smoothing was applied to

the available wave spectra to select interpolated peaks at a higher-frequency resolution. The resulting values of $Tp$ and $Hs$ are shown in the heat map scatter data of Fig. 9. Extreme $Tp$ and $Hs$ contours were generated for the relevant return periods for the ultimate load cases. The principal component analysis method within MHKit was used (Klise et al., 2020)̇This fit has been shown to better represent measured data of extreme events (Eckert-Gallup et al., 2016). The principal component analysis contours are shown in Fig. 9 for LC 6.1, 6.3, and 6.5. The $H_s$-dependent $Tp$ ranges for the fatigue load case are not based on

extreme value theory but mark the 0.01 and 0.99 quantiles of the NBDC data. The range of peak periods is a function of $Hs$ selected for each run; the bounds of the range are the intersections of the relevant contour and $Hs$.





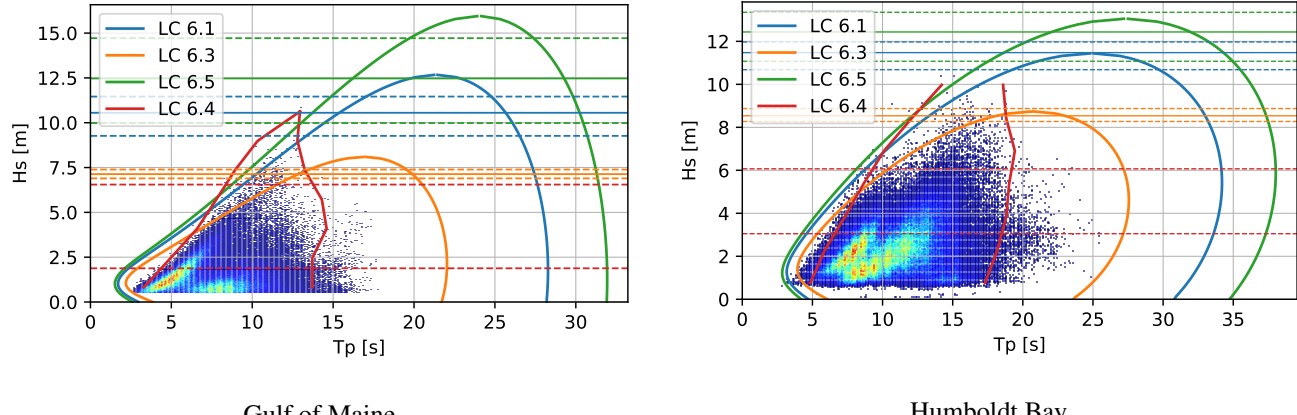

Gulf of Maine                                              Humboldt Bay

**Figure 9.** $Tp$ and $Hs$ contours with a return period of 50, 1, and 500 years corresponding to IEC design load cases 6.1, 6.3, and 6.5, respectively, for the Gulf of Maine (left) and Humboldt Bay (right). Extreme contours are based on the principal component analysis method, and the normal sea state contour for LC 6.4 is based on the 0.01 and 0.99 quantile of the NBDC data shown in the underlying heat map. Horizontal lines show the $Hs$ ranges and nominal values for each load case.

The wave spectral shape factor, $\gamma_{wave}$, has a range as recommended by the American Bureau of Shipping (ABS, 2016). The nominal value is a function of $Hs$ and $Tp$ as specified by IEC (IEC, 2024).

Current data also came from NDBC; station 46022 was again used for Humboldt Bay, but station 44032 was used for the
285 Gulf of Maine current data. The data from 44032 included a full depth profile, whereas the 46022 data only included one depth. At station 44032, a depth of 14.0 m was selected. The GEV bootstrapping method was used to determine the extreme value uncertainty ranges. The resulting distributions and limits are shown in Fig. 10. The range of speeds is much more narrow for Humboldt Bay, and the difference between values with increasing return periods is also much smaller. The distribution for the Gulf of Maine is more positively skewed, resulting in increasingly large values with a large return period. The upper limit of
290 the 500-year $V_{current}$ in the Gulf of Maine was deemed unrealistic and was capped at 3.0 m/s. Current direction is allowed to range from in line with the wind direction to fully opposing.





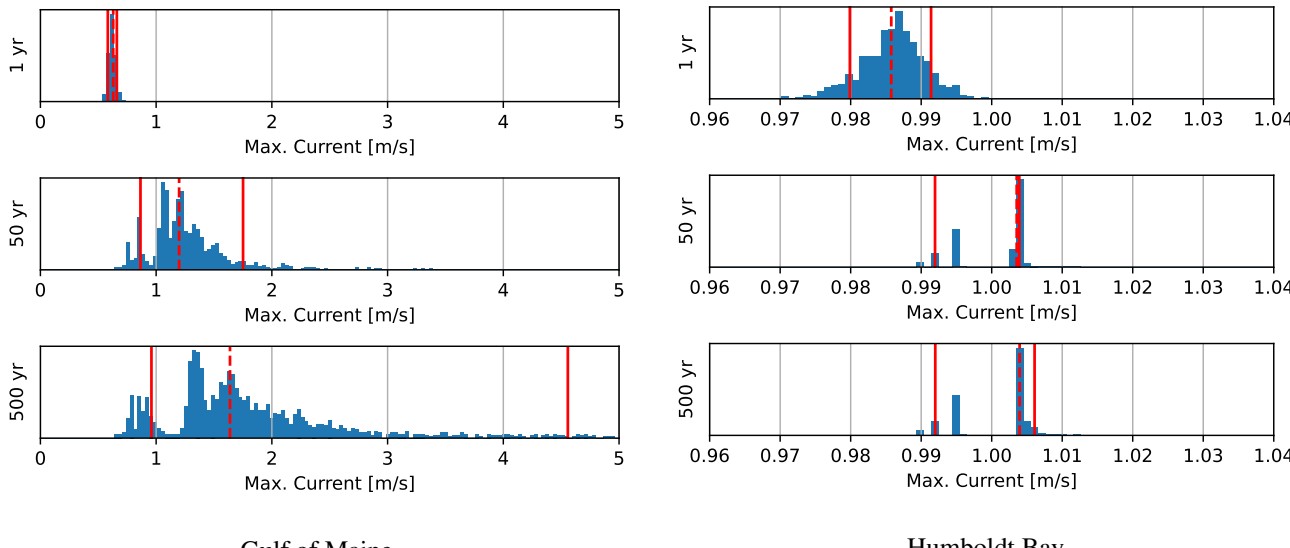

Gulf of Maine

Humboldt Bay

**Figure 10.** Bootstrapped distribution of $V_{current}$ with return periods of 1, 50, and 500 years based on a generalized extreme value distribution for the Gulf of Maine (left) and Humboldt Bay (right). The nominal value is based on the full dataset with no resampling and is shown with a dashed red line. The limits of the parameter range are the 0.1 and 0.9 quantiles of the bootstrapped distribution and are shown with solid red lines

Nominal values for the platform viscous drag coefficients came from tuning efforts to best match tank test data and CFD simulations for the UMaine VolturnUS-S platform published in 2023 (Fowler et al., 2023). The ranges for the cylindrical transverse coefficients are based on the range of expected Reynolds numbers and Keulegan–Carpenter numbers. The selected

ranges match those of a sensitivity study performed for this platform by a group from CENER and University of Stuttgart that focused specifically on platform drag (Sandua-Fernandez et al., 2022). Both $Cd_{axial}$ and $Cd_{rectangular}$ have viscous drag due to separated flow around sharp corners. The ranges are based on Keulegan–Carpenter number dependence found in experiments done by a group from the Universities of Edinburgh, Strathclyde, and Glasgow and Fyvie with horizontally submerged rectangular cylinders (Venugopal et al., 2008).

**4  Output Quantities of Interest**

The simulated fatigue and extreme loads are evaluated for 12 QOI as shown in Table 5. LC 6.5, with extreme environmental conditions with a return period of 500 years, is only meant to be used to evaluate the support structure, so only motion and loads in the support structure are considered. The ultimate load sensitivity of the remaining seven QOI are only assessed for LC 6.1 and 6.3.





**Table 5.** Output quantities of interest and the load cases they are evaluated for

|  | Ultimate Only | | | Fatigue Only |
| --- | --- | --- | --- | --- |
|  | LC 6.1 | LC 6.3 | LC 6.5 | LC 6.4 |
| Nacelle Acceleration | X | X |  | X |
| Heel Angle | X | X | X | X |
| Watch Circle | X | X | X | X |
| Anchor Tension | X | X | X | X |
| Fairlead Tension | X | X | X | X |
| Blade Tip Deflection | X | X |  | X |
| Twr. Base Mom. | X | X | X | X |
| Yaw-bearing Yaw Mom. | X | X |  | X |
| Yaw-bearing Bending Mom. | X | X |  | X |
| LS Shaft Bending Mom. | X | X |  | X |
| Root Pitching Mom. | X | X |  | X |
| Root Bending Mom. | X | X |  | X |

The blade root bending moment, yaw-bearing bending moment, tower base bending moment, low-speed shaft bending moment, and watch circle are each a composite of components in two directions. For ultimate loads, the maximum vector magnitude was selected in any direction. For fatigue loads, cycles in different directions were not considered together; instead, the vectors were divided into 12 directional bins, and the bin with the highest fatigue value was used for the QOI.

    Nacelle acceleration, which is also a multidirectional vector, was taken as the magnitude. Heel angle was evaluated as the

inverse tangent of the vector magnitude of the tangents of pitch and roll. For both ultimate and fatigue loads, the mooring line with the largest load was used for the QOI.

## 5   Results

A total of 209,920 OpenFAST and 46,080 TurbSim runs were used in the following analysis. The total number comes from the product of 2 locations, 4 load cases, 32 starting points, 20 seed numbers, and (1+40) input perturbations for OpenFAST or (1+8)

wind input perturbations for TurbSim. If not otherwise specified, all results include the outputs from all relevant simulations.

    All simulations were postprocessed for the relevant ultimate or fatigue EE values, and significant EE events were noted. Figures A1 and A2 in Appendix A show histograms for QOI values from all ultimate LC runs and the thresholds for significance.

    Figure 11 shows the number of significant ultimate EE events for the Gulf of Maine. Each bar along the horizontal axis is one of the 40 variable input parameters. Each bar is broken and colored according to the QOI that experienced a significant change

with the input perturbation. QOI bars with hash marks pertain to global system loads while QOI bars without hash marks are rotor-specific. None of the wind input parameters has a dominant sensitivity; this dramatically differs from previous analyses with operational load cases, where $\sigma_u$ was the primary sensitivity for the majority of QOI (Wiley et al., 2023). The three inputs





with the most significant EE values are all direction-based: $\Theta_{current}$, $Yaw$, and $\Theta_{mis}$. The $V_{current}$ also contributes a large number of significant events. Both current parameters have a large impact on the watch circle and drive fairlead and mooring

loads as well. It is interesting that the direction of the waves relative to the wind, $\Theta_{mis}$, and the relative direction of the wind to the turbine, $Yaw$, have such an important impact, but the input parameters describing the height and period of the waves have such little impact. The $COG_X$ also has significant impacts; this appears to be largely driven by the effect on the heel angle and loads impacted by the heel angle.

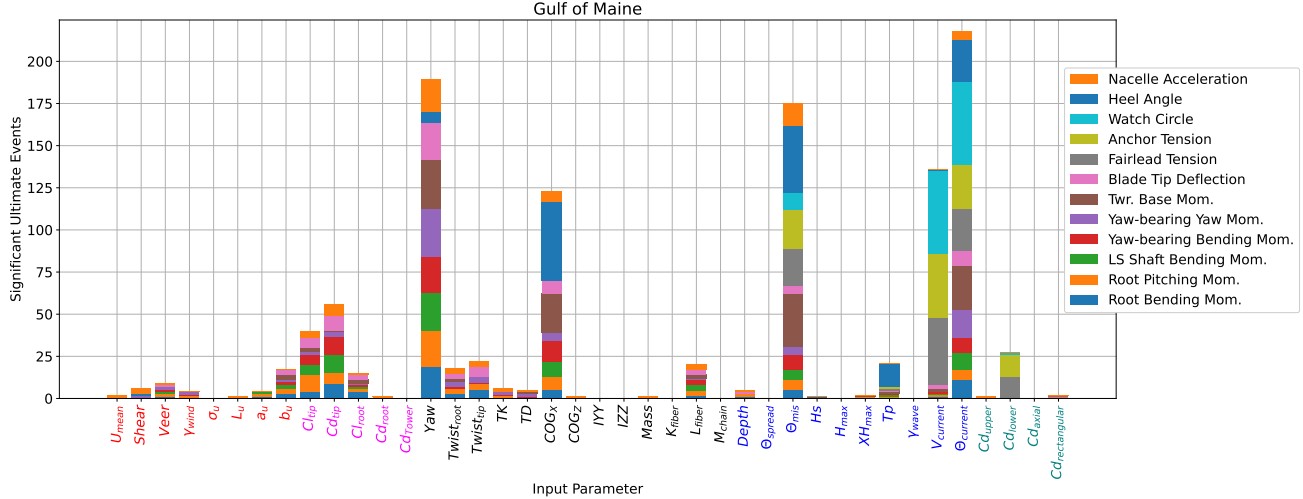

**Figure 11.** Sensitivity of ultimate loads EE values for the IEA 15 MW RWT on the VolturnUS-S platform in the Gulf of Maine

Figure 12 shows the ultimate EE sensitivities for Humboldt Bay. Similar to the results in the Gulf of Maine, $\Theta_{current}$, $\Theta_{mis}$,

$COG_X$, and $Yaw$ drive large sensitivities. Different from the Gulf of Maine, $V_{current}$ has no associated significant events. This is most likely because of the very tight distribution of current speeds in Humboldt Bay, leading to less uncertainty in the extreme values, and a smaller range. The most dominant input parameter, mostly driving global QOI like the watch circle and mooring loads, is the $L_{fiber}$. The taut mooring system in the deeper Humboldt Bay has much longer sections of polyester in the mooring design. This indicates that installations with similar mooring designs need to have very thorough analysis and

monitoring of creep.

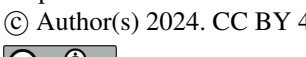


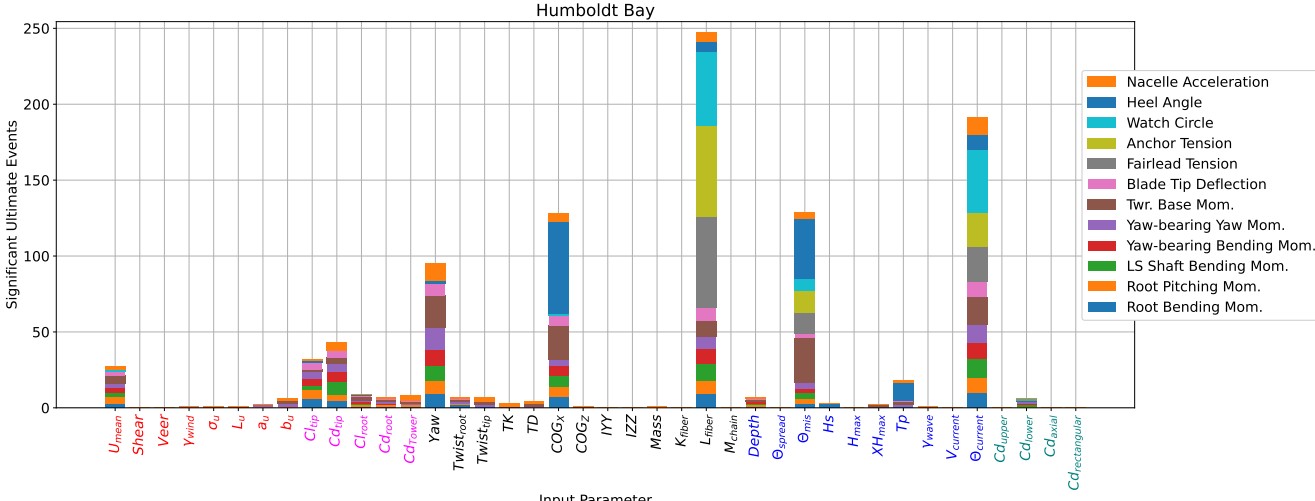

**Figure 12.** Sensitivity of ultimate loads EE values for the IEA 15 MW RWT on the VolturnUS-S platform in Humboldt Bay

Figures 11 and 12 include runs from LC 6.1, 6.3, and 6.5. Figure 13 includes the same sensitivities but calculated only with each load case individually for the Gulf of Maine. The threshold for a significant event is recalculated based only on the runs from each load case. For comparison to the aggregate results, the number of significant events are normalized so that the sum for each load case is 1.0. Only the input parameters with a nonzero number of significant events are included. The most

dominant input parameters are similar when considering the individual load cases but display some differences. For example, $Yaw$ is clearly the primary QOI driver for LC 6.3, which includes the extreme range of $Yaw$. The relatively large impact of $Yaw$ masks some of the other important sensitivities in LC 6.3, including $V_{current}$. $Yaw$ is still important without the extreme range check, with secondary importance for LC 6.1 with a smaller range of angles. Sensitivity to $V_{current}$ is large for LC 6.5, where the 500-year return period has a large range. Even with a relatively large range of the 500-year $Hs$ in LC 6.5, there is

almost no significant impact caused by the wave height, while the wave direction, $\Theta_{mis}$, is a primary parameter for all three load cases. The size and position of the embedded $H_{max}$ similarly have little to no significant impact.



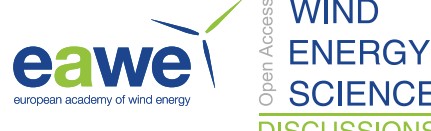

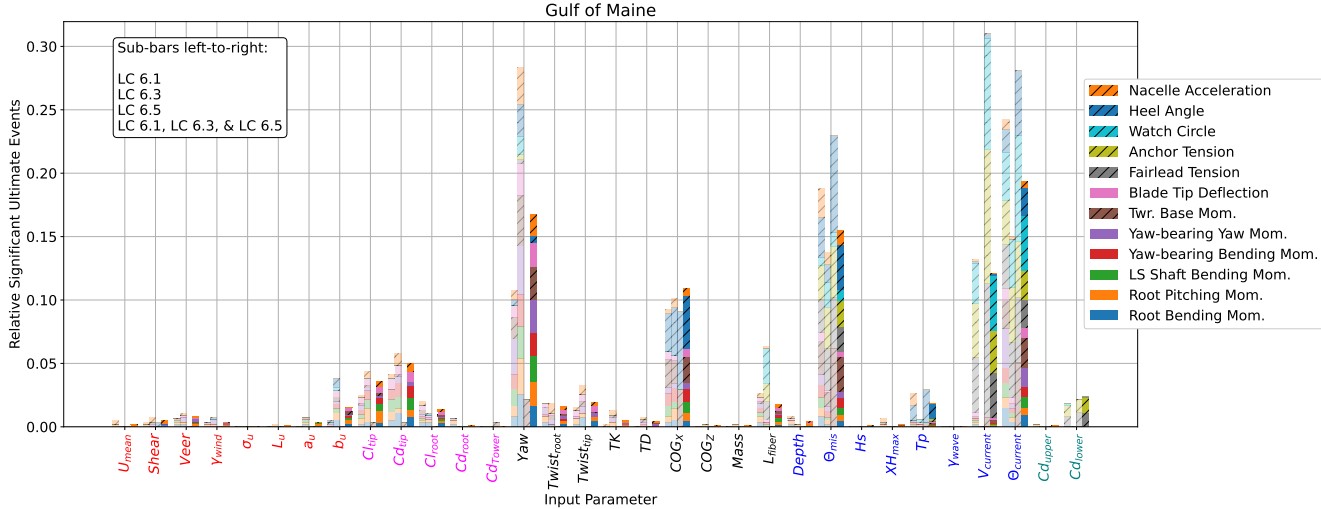

**Figure 13.** Sensitivity of ultimate loads EE values for the IEA 15 MW RWT on the VolturnUS-S platform in the Gulf of Maine based on IEC LC 6.1, 6.3, 6.5 individually and on all three together. Sensitivities based on individual load cases are shown partially transparent, and the sensitivities from the aggregate are opaque.

Figure 14 shows the same individual load case results for Humboldt Bay. Again, the primary sensitivities are similar to when all load cases are used. $Yaw$ has many fewer associated significant events when only LC 6.5 is considered. In this load case, the mooring $L_{fiber}$ has a dominant effect.





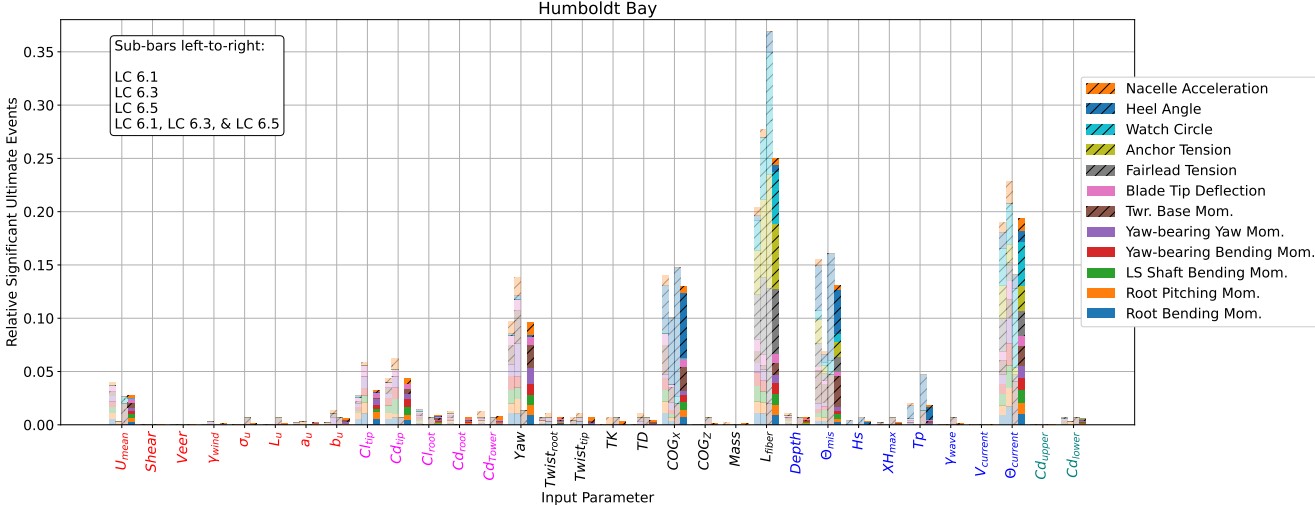

**Figure 14.** Sensitivity of ultimate loads EE values for the IEA 15 MW RWT on the VolturnUS-S platform in Humboldt Bay based on IEC LC 6.1, 6.3, 6.5 individually and on all three together. Sensitivities based on individual load cases are shown partially transparent, and the sensitivities from the aggregate are opaque.

The fatigue sensitivities are based fully on LC 6.4 and are shown in Fig. 15 for the Gulf of Maine. The fatigue loads are most strongly influenced by the waves. $\Theta_{mis}$, $Hs$, and $Tp$ contribute the most significant events. Unlike with ultimate loads, the height and period matter in addition to the direction of the waves. None of the $Tp$ ranges overlap with the resonant periods of UMaine VolturnUS-S, but the difference in fatigue response is still significant. $Veer$, again pertaining to the load directionality, is the most influential incident wind parameter. This is in contrast to a previous EE sensitivity analysis of a floating wind system in operational conditions, where $Veer$ was found to be the incident wind parameter with the least influence (Wiley et al., 2023).





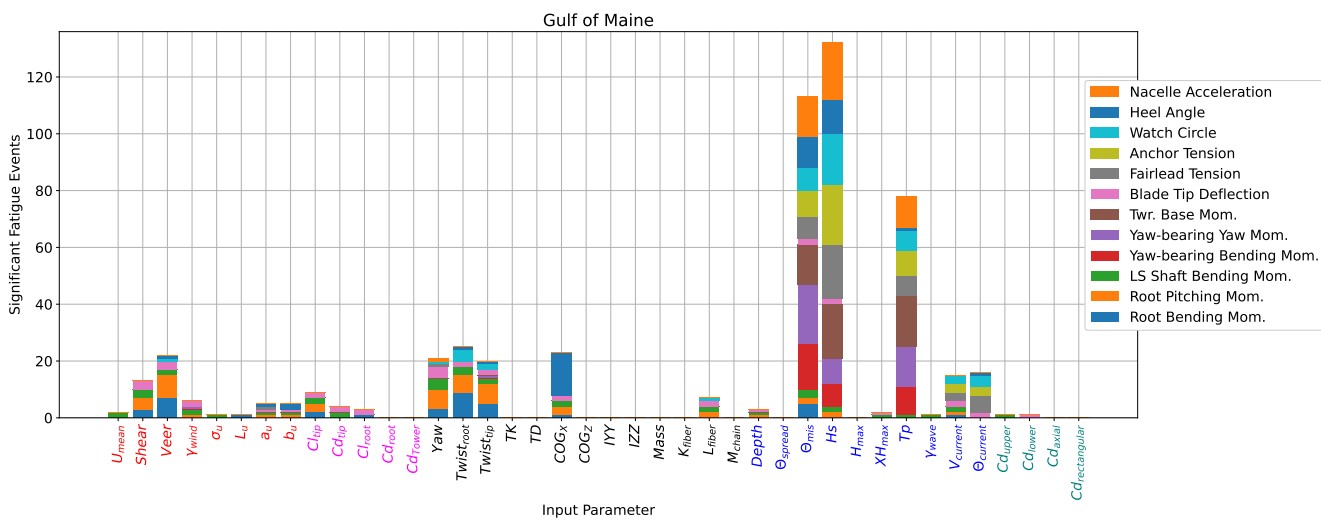

**Figure 15.** Sensitivity of fatigue loads EE values for the IEA 15 MW RWT on the VolturnUS-S platform in the Gulf of Maine

Figure 15 shows the fatigue results for Humboldt Bay. The same three wave parameters are still important, but the relative influence of $Hs$ is smaller. Similar to the ultimate loads, $L_{fiber}$ contributes many significant EE events. Not only does the length of the polyester in the taut system impact mean displacements, but the change to the mooring stiffness has important effects on the platform oscillations. Similarly, $COG_X$ not only causes a mean heel offset but also impacts cyclic loads.

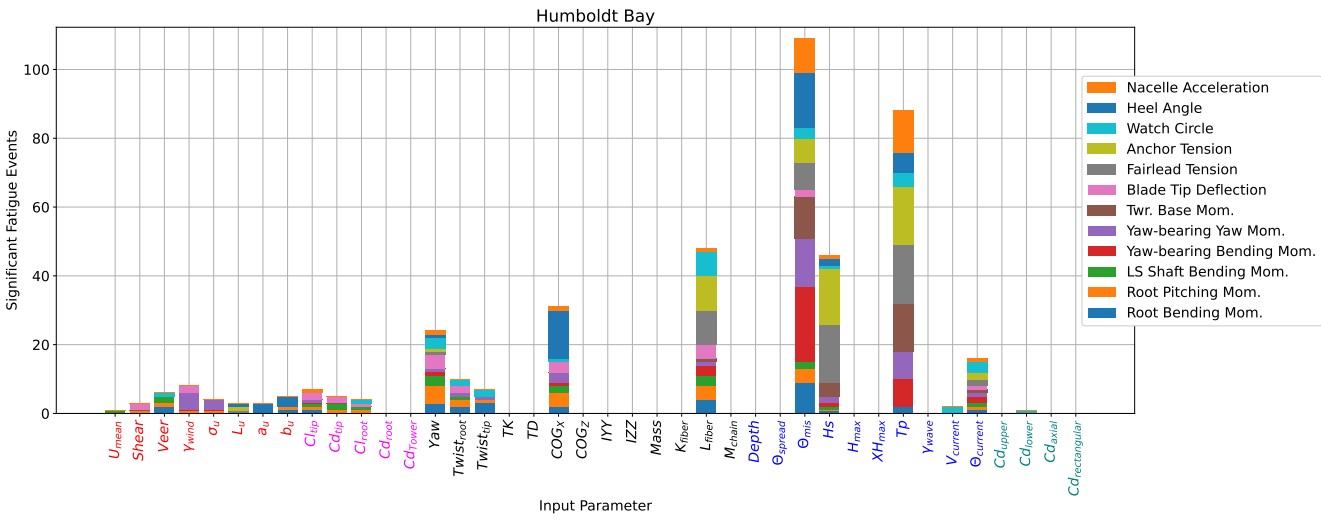

**Figure 16.** Sensitivity of fatigue loads EE values for the IEA 15 MW RWT on the VolturnUS-S platform in Humboldt Bay



Exact fatigue failure stress curves for each system component are not used for the load quantification described in Eq. (2). Figures C1 and C2 in Appendix C show the resulting fatigue sensitivities if the standard deviation was used as a proxy for fatigue instead. There are no significant changes in the primary sensitivities.

Directionality appears to drive many of the idling condition loads for this floating system. $\Theta_{mis}$ is particularly interesting. The run setup in this study treats the wind direction as a constant, directly down the line of one of the mooring lines and

rectangular members. Misalignment from this angle is defined by $Yaw$ for the rotor, $\Theta_{mis}$ for the waves, and $\Theta_{current}$ for the current. The input parameter definition does not make it clear whether sensitivity to $\Theta_{mis}$ is because of the relative difference between the wind and wave heading, or because of the difference between the platform and wave heading. The wave loading, particularly the viscous drag effects on the rectangular members, changes depending on the relative angle of the flow and the platform.

A single additional simulation was run for the nominal starting point of LC 6.1 in the Gulf of Maine. A perturbation was made collectively to the wave heading and the wind direction as shown in the graphic in the bottom right of Fig. 17. The value of the perturbation is the same as originally tested for only the wave heading, shown in the middle graphic. Each QOI is shown for these two runs as a fraction of the QOI at the nominal starting point. If there is a deviation from the starting point with a wave-only heading change, but not with a collective heading change, the impact likely comes from the misalignment

of the wind and wave loading. If the deviation is similar, the impact likely comes from the differences in wave loading on the platform. It is difficult to draw conclusions from a single run comparison, but it appears that the wave orientation to the platform may be more important or equally important as the wave orientation to the wind. Separating the wave heading from $\Theta_{mis}$ should be investigated in future work with a full investigation of the parameter hyperspace.





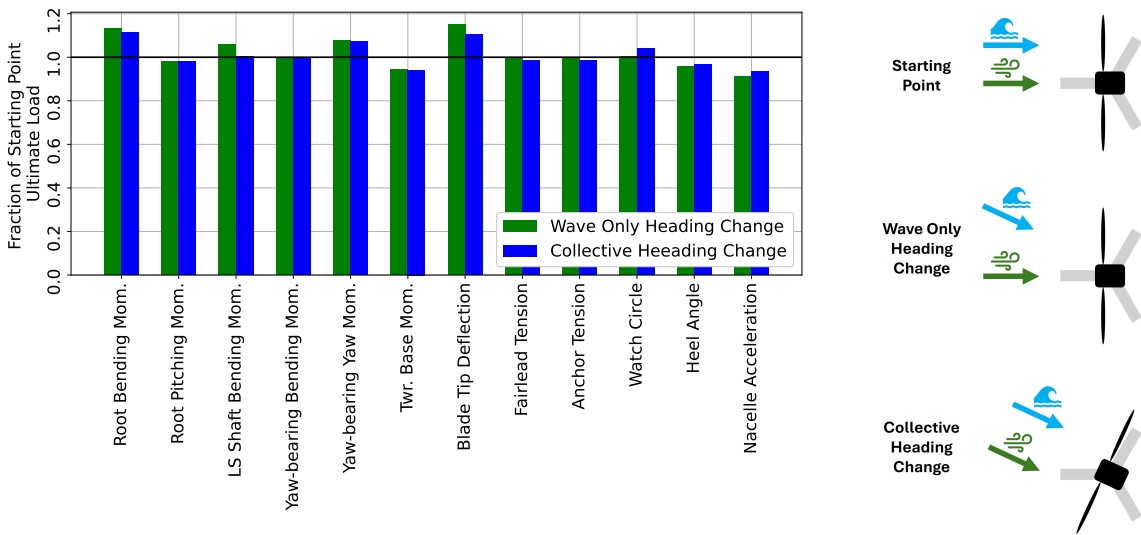

**Figure 17.** Changes to output loads with a perturbation in wave heading only ($\theta_{mis}$) compared to a perturbation in collective wave heading, inflow wind angle, and nacelle yaw for the IEA 15 MW RWT on the VolturnUS-S platform in the Gulf of Maine using nominal values for all input parameters as the starting point

Tables B1, B2, B3, and B4 in Appendix B display the results for individual input parameter and QOI combination sensitivities in a detailed format. The fraction of possible runs (the total number of perturbations for a given input parameter) that exceed the significance threshold for a given QOI are shown with coloration to highlight the dominant sensitivities. A value of 1.0 would mean that all perturbations for that input parameter resulted in a significant event for that QOI.

## 6 Seed Convergence

The turbulent wind and irregular wave fields rely on a random number generator to assign phases. The seed number can be dictated to have reproducible results and ensure unique environments if it is changed. There can be variations in loads due to differences in seed number. For an EE sensitivity analysis, a sufficient number of seed numbers needs to be used to clearly differentiate changes due to input parameter perturbations from changes in seed number. The QOI calculations in Eqs. (1) and (2) average across all seed numbers run for a certain set of input parameters. When enough seed numbers are run, the results will converge so that additional seeds/runs do not result in important changes to the results.

Figures D3 and D4 show the convergence of one QOI for one starting point and one perturbation. Given the large number of input parameters, QOI, and load cases, it would be very difficult to track the convergence for each combination of input and output. Figure 18 shows how the final ultimate EE sensitivities would look across all relevant runs, if an increasing quantity of seed numbers were used, using the results from all 32 starting points. Ultimately, the relative importance of input parameters




throughout the parameter hyperspace is the desired information. No changes to the primary or secondary sensitivities appear
to occur after 15 seed numbers are used.

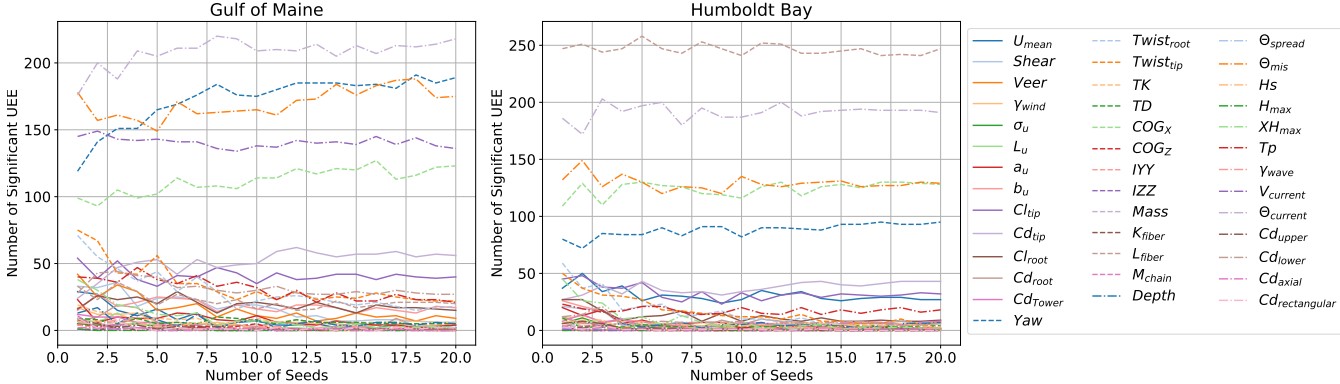

**Figure 18.** Convergence of ultimate load EE sensitivity for the IEA 15 MW RWT on the VolturnUS-S platform with an increasing number
of random seed numbers

Figure 19 shows this same convergence for the fatigue EE sensitivities. Again, the relative importance does not appear to
change after 15 seed numbers are used. Twenty seed numbers were used for each input parameter combination in this study.

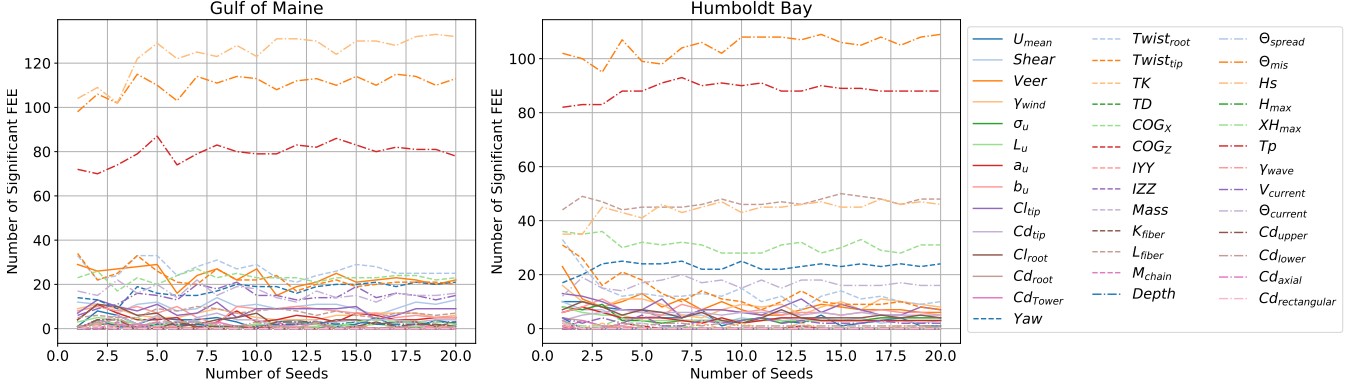

**Figure 19.** Convergence of fatigue load EE sensitivity for the IEA 15 MW RWT on the VolturnUS-S platform with an increasing number of
random seed numbers

## 7  Starting Point Convergence

The results from perturbations around each starting point provide information about the local sensitivities at that location in
the parameter hyperspace. As the number of starting points increases and the aggregate results are analyzed, the resulting
sensitivities approach the global sensitivity values. It is possible that at one point in the parameter hyperspace there is a very





strong sensitivity to some input parameter that is not felt with other combinations of inputs. Many of the input parameters have highly coupled effects, potentially leading to unique local sensitivities. To combine the impacts of these couplings across the full range of all parameters, enough starting points need to be used.

Similar to the demonstration of seed convergence, Fig. 20 shows what the final relative sensitivities would be for ultimate loads if an increasing number of starting points were used (results from all 20 seed numbers are used). This convergence is tracked with additional starting points based on the Sobol sequence, which should evenly fill the parameter hyperspace, to systematically approach the global sensitivity with as few points as possible. Note the discontinuity in the Humboldt Bay results with the addition of starting point 18. QOI with this combination of variables happened to be uniquely sensitive to $L_{fiber}$ and

$\Theta_{current}$, resulting in a lasting change to the relative significance. It is possible that unique coupled impacts like this occur for untested combinations of inputs; however, it appears that the identification of the dominant ultimate EE sensitivities has converged after 30 starting points are used. A total of 32 starting points were simulated in this analysis.

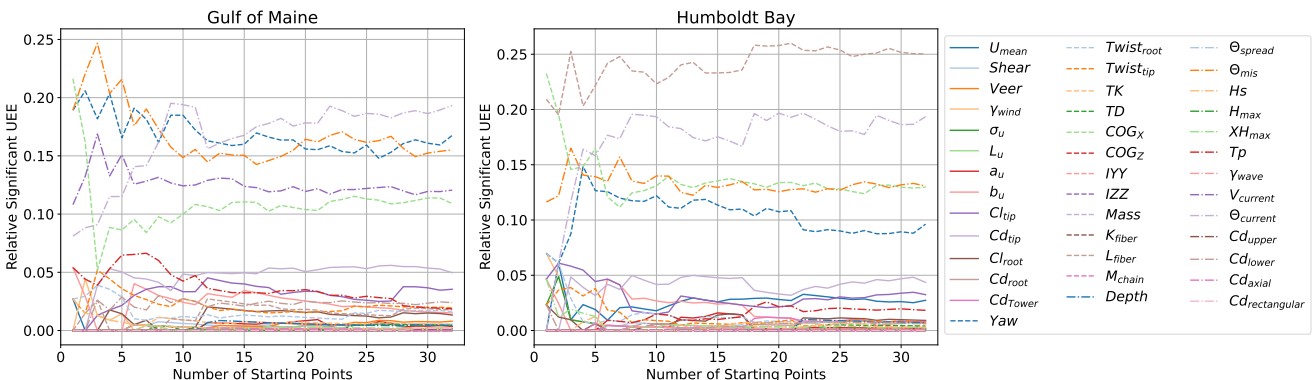

**Figure 20.** Convergence of ultimate load EE sensitivity for the IEA 15 MW RWT on the VolturnUS-S platform with an increasing number of EE starting points

Figure 21 shows the starting point convergence for the fatigue EE sensitivity. No significant changes occur past 25 starting points.

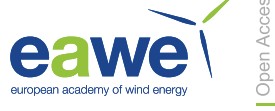
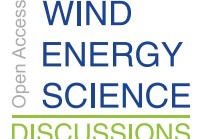


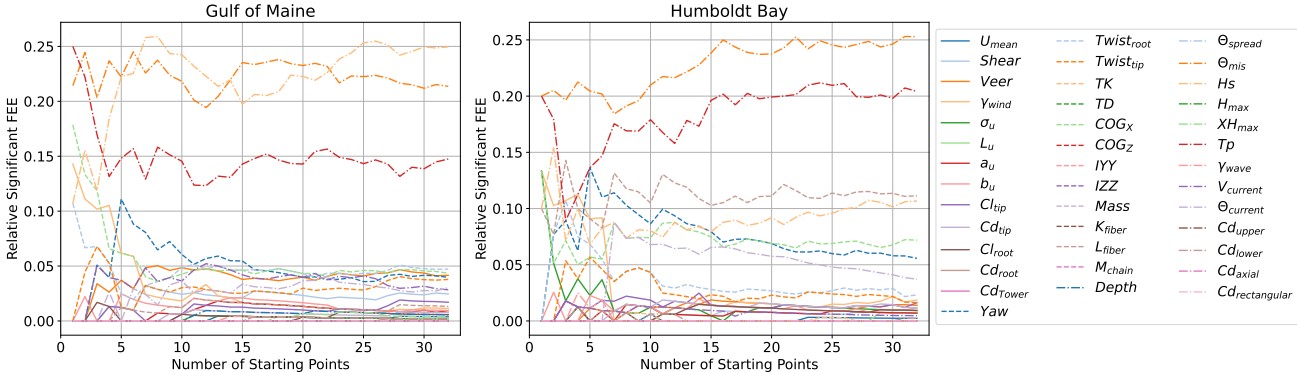

**Figure 21.** Convergence of fatigue load EE sensitivity for the IEA 15 MW RWT on the VolturnUS-S platform with an increasing number of EE starting points

## 8   Conclusions

An EE approach was effectively used to identify the input parameter ranges that can have the largest impacts on ultimate and fatigue loads, specifically for above-cut-out wind conditions with an idling rotor. The dynamic stall effects and nonlinear large blade deflections were not accurately captured in the efficient modeling approach used. Blade tip deflections, in particular, would likely be affected by these missing physical phenomena. Interestingly, the distribution of blade tip deflections was generally narrow enough such that a small number of significant events came from this QOI. This indicates that for the load cases studied here, uncertainties in other aspects of the numerical model, which are more accurately captured in the chosen models, are likely more important.

While wave parameters were found to have minimal impacts on the loads of an operating floating wind turbine system in previous work, the direction of waves was found to make large differences in global ultimate loads, and the direction, period, and height were found to drive global fatigue loads. Incident wind input parameters had a small impact on both ultimate and fatigue loads – again, a large contrast to operational load case results (Wiley et al., 2023). For extreme environmental load cases investigated in this study, more attention needs to be given to the uncertainty in wave and current parameters compared to wind parameters. This is expected due to a significant drop in the wind loading for idling conditions.

Directionality in general was shown to make large differences in the loads experienced by the IEA 15 MW UMaine VolturnUS-S system. The orientation of the rotor, waves, and current were all primary ultimate sensitivities, and the wave direction had a strong influence on fatigue loads. There is some ambiguity as to whether the relative misalignment of the waves to the wind or the waves to the platform is more important, but it appears that both angles matter. The importance of direction could be due to a nonuniform mooring stiffness or due to differences in excitation and damping in different directions.



Variations within the range of mooring system properties did not have dominant effects on the loads in the semi-taut moored
Gulf of Maine system. However, for the taut Humboldt Bay system, with long polyester segments, the $L_{fiber}$ is influential. As
more deep-water systems are designed, focus should be placed on modeling and monitoring fiber line creep.

*Code availability.* The base OpenFAST input files will be publicly available, most likely on Zenodo (FINALIZE LOCATION BEFORE
FINAL VERSION).

## Appendix A:  Detailed Ultimate EE Values

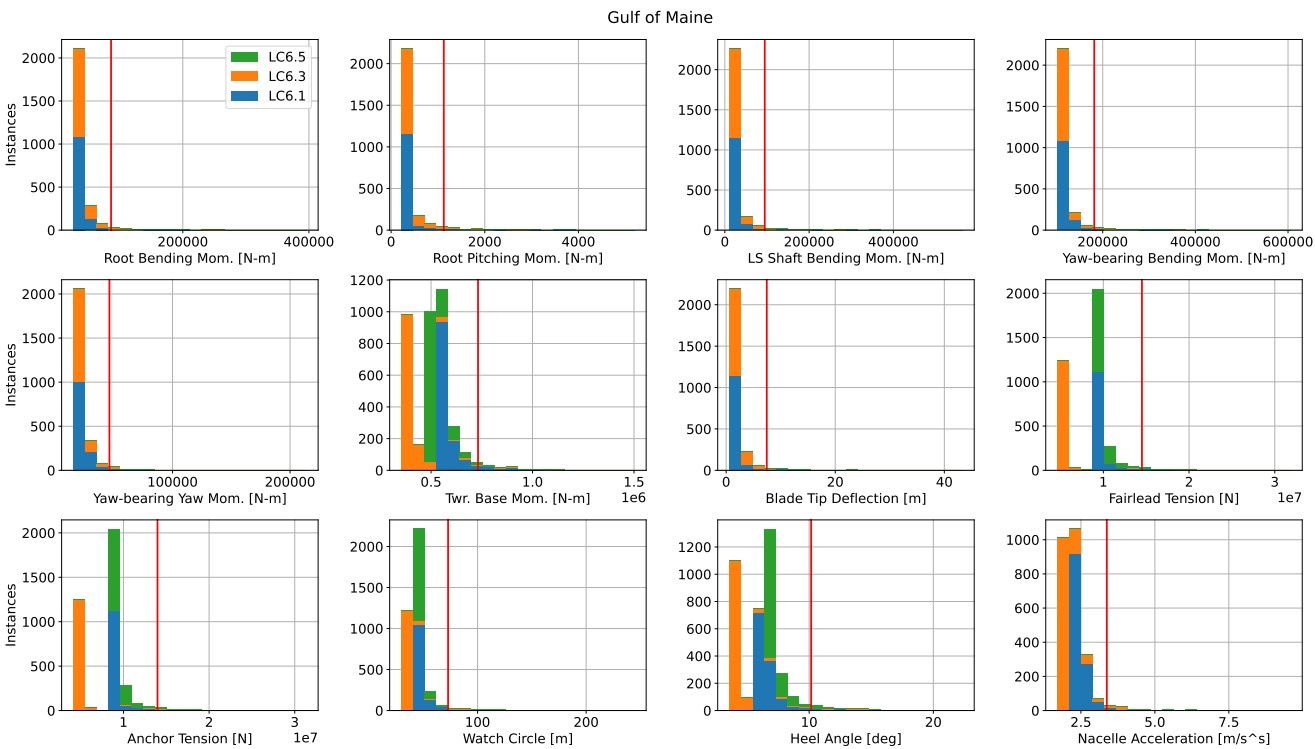

**Figure A1.** Histogram of ultimate EE values for the IEA 15 MW RWT on the VolturnUS-S platform in the Gulf of Maine for IEC design LC
6.1, 6.3, and 6.5. Red line indicates two standard deviations above the mean, which is the threshold for a significant event.

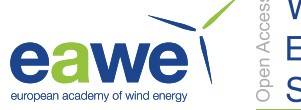
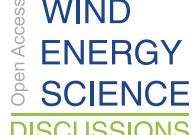

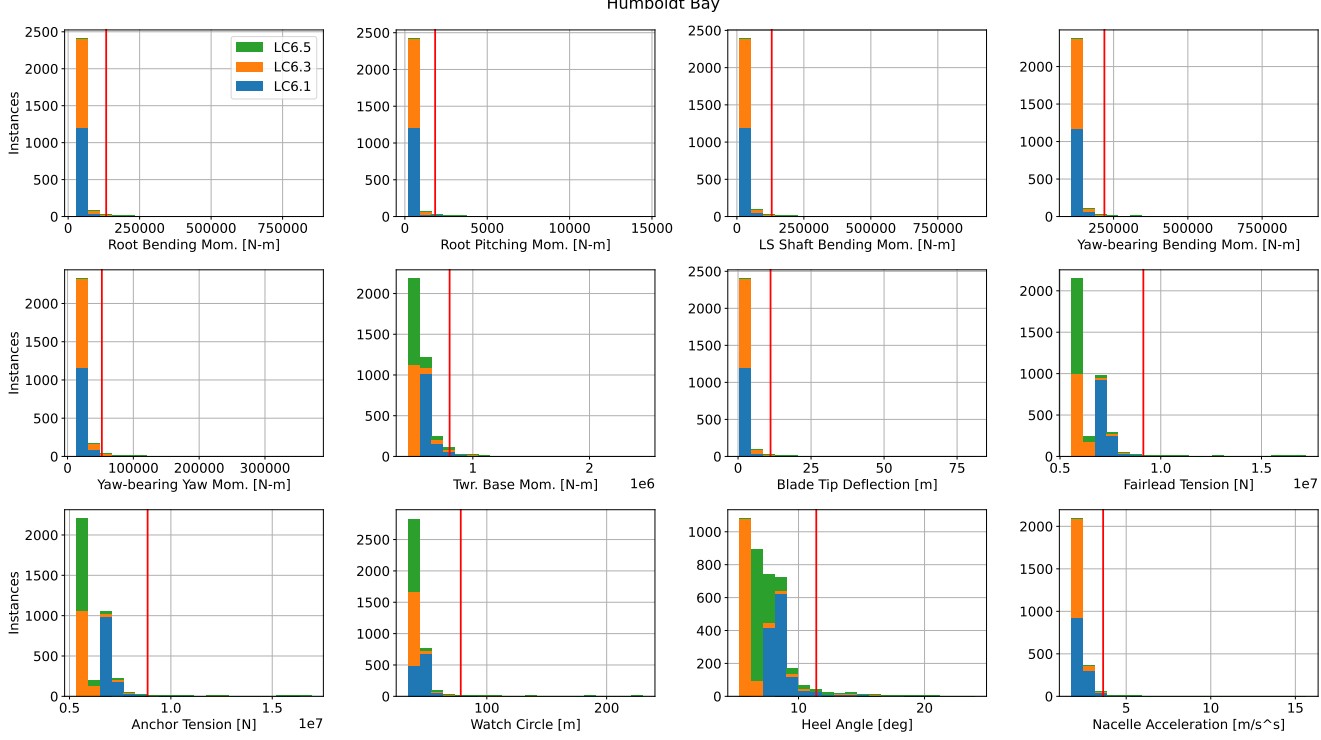

**Figure A2.** Histogram of ultimate EE values for the IEA 15 MW RWT on the VolturnUS-S platform in Humboldt Bay for IEC design LC 6.1, 6.3, and 6.5. Red line indicates two standard deviations above the mean, which is the threshold for a significant event.





## 440 Appendix B: Sensitivity Tables

| | Root Bending Mom. | Root Pitching Mom. | LS Shaft Bending Mom. | Yaw-bearing Bending Mom. | Yaw-bearing Yaw Mom. | Twr. Base Mom. | Blade Tip Deflection | Fairlead Tension | Anchor Tension | Watch Circle | Heel Angle | Nacelle Acceleration |
|---|---|---|---|---|---|---|---|---|---|---|---|---|
| $U_{mean}$ | 0.00 | 0.03 | 0.00 | 0.00 | 0.00 | 0.00 | 0.00 | 0.00 | 0.00 | 0.00 | 0.00 | 0.00 |
| $Shear$ | 0.00 | 0.00 | 0.00 | 0.00 | 0.03 | 0.00 | 0.00 | 0.00 | 0.00 | 0.00 | 0.01 | 0.05 |
| $Veer$ | 0.02 | 0.03 | 0.02 | 0.02 | 0.03 | 0.00 | 0.02 | 0.00 | 0.00 | 0.00 | 0.00 | 0.02 |
| $\gamma_{wind}$ | 0.00 | 0.03 | 0.00 | 0.00 | 0.03 | 0.00 | 0.00 | 0.00 | 0.00 | 0.00 | 0.00 | 0.00 |
| $\sigma_u$ | 0.00 | 0.00 | 0.00 | 0.00 | 0.00 | 0.00 | 0.00 | 0.00 | 0.00 | 0.00 | 0.00 | 0.00 |
| $L_u$ | 0.00 | 0.00 | 0.00 | 0.00 | 0.00 | 0.00 | 0.00 | 0.00 | 0.00 | 0.00 | 0.00 | 0.02 |
| $a_u$ | 0.02 | 0.03 | 0.02 | 0.00 | 0.00 | 0.00 | 0.00 | 0.00 | 0.00 | 0.00 | 0.00 | 0.00 |
| $b_u$ | 0.05 | 0.05 | 0.03 | 0.03 | 0.02 | 0.03 | 0.05 | 0.00 | 0.00 | 0.00 | 0.00 | 0.00 |
| $Cl_{tip}$ | 0.06 | 0.16 | 0.09 | 0.09 | 0.03 | 0.02 | 0.09 | 0.00 | 0.00 | 0.00 | 0.00 | 0.06 |
| $Cd_{tip}$ | 0.14 | 0.09 | 0.17 | 0.17 | 0.05 | 0.00 | 0.14 | 0.00 | 0.00 | 0.00 | 0.00 | 0.11 |
| $Cl_{root}$ | 0.06 | 0.03 | 0.02 | 0.02 | 0.02 | 0.02 | 0.05 | 0.00 | 0.00 | 0.00 | 0.00 | 0.02 |
| $Cd_{root}$ | 0.00 | 0.02 | 0.00 | 0.00 | 0.00 | 0.00 | 0.00 | 0.00 | 0.00 | 0.00 | 0.00 | 0.00 |
| $Cd_{Tower}$ | 0.00 | 0.00 | 0.00 | 0.00 | 0.00 | 0.00 | 0.00 | 0.00 | 0.00 | 0.00 | 0.00 | 0.00 |
| $Yaw$ | 0.30 | 0.33 | 0.36 | 0.33 | 0.45 | 0.30 | 0.34 | 0.00 | 0.00 | 0.00 | 0.06 | 0.30 |
| $Twist_{root}$ | 0.05 | 0.05 | 0.00 | 0.02 | 0.05 | 0.02 | 0.05 | 0.00 | 0.00 | 0.00 | 0.00 | 0.05 |
| $Twist_{tip}$ | 0.08 | 0.06 | 0.00 | 0.00 | 0.06 | 0.00 | 0.09 | 0.00 | 0.00 | 0.00 | 0.00 | 0.05 |
| $TK$ | 0.00 | 0.03 | 0.00 | 0.00 | 0.03 | 0.00 | 0.00 | 0.00 | 0.00 | 0.00 | 0.00 | 0.03 |
| $TD$ | 0.00 | 0.00 | 0.00 | 0.02 | 0.03 | 0.01 | 0.00 | 0.00 | 0.00 | 0.00 | 0.00 | 0.02 |
| $COG_X$ | 0.08 | 0.12 | 0.14 | 0.19 | 0.08 | 0.24 | 0.12 | 0.00 | 0.00 | 0.00 | 0.49 | 0.09 |
| $COG_Z$ | 0.00 | 0.02 | 0.00 | 0.00 | 0.00 | 0.00 | 0.00 | 0.00 | 0.00 | 0.00 | 0.00 | 0.00 |
| $IYY$ | 0.00 | 0.00 | 0.00 | 0.00 | 0.00 | 0.00 | 0.00 | 0.00 | 0.00 | 0.00 | 0.00 | 0.00 |
| $IZZ$ | 0.00 | 0.00 | 0.00 | 0.00 | 0.00 | 0.00 | 0.00 | 0.00 | 0.00 | 0.00 | 0.00 | 0.00 |
| $Mass$ | 0.00 | 0.02 | 0.00 | 0.00 | 0.00 | 0.00 | 0.00 | 0.00 | 0.00 | 0.00 | 0.00 | 0.00 |
| $K_{fiber}$ | 0.00 | 0.00 | 0.00 | 0.00 | 0.00 | 0.00 | 0.00 | 0.00 | 0.00 | 0.00 | 0.00 | 0.00 |
| $L_{fiber}$ | 0.03 | 0.05 | 0.05 | 0.05 | 0.02 | 0.02 | 0.05 | 0.00 | 0.00 | 0.00 | 0.00 | 0.05 |
| $M_{chain}$ | 0.00 | 0.00 | 0.00 | 0.00 | 0.00 | 0.00 | 0.00 | 0.00 | 0.00 | 0.00 | 0.00 | 0.00 |
| $Depth$ | 0.02 | 0.03 | 0.00 | 0.00 | 0.00 | 0.00 | 0.02 | 0.00 | 0.00 | 0.00 | 0.00 | 0.02 |
| $\Theta_{spread}$ | 0.00 | 0.00 | 0.00 | 0.00 | 0.00 | 0.00 | 0.00 | 0.00 | 0.00 | 0.00 | 0.00 | 0.00 |
| $\Theta_{mis}$ | 0.08 | 0.09 | 0.09 | 0.14 | 0.08 | 0.32 | 0.08 | 0.23 | 0.24 | 0.10 | 0.42 | 0.20 |
| $Hs$ | 0.00 | 0.00 | 0.00 | 0.00 | 0.00 | 0.00 | 0.00 | 0.00 | 0.00 | 0.01 | 0.00 | 0.00 |
| $H_{max}$ | 0.00 | 0.00 | 0.00 | 0.00 | 0.00 | 0.00 | 0.00 | 0.00 | 0.00 | 0.00 | 0.00 | 0.00 |
| $XH_{max}$ | 0.00 | 0.02 | 0.00 | 0.00 | 0.00 | 0.01 | 0.00 | 0.00 | 0.00 | 0.00 | 0.00 | 0.00 |
| $Tp$ | 0.00 | 0.02 | 0.02 | 0.02 | 0.00 | 0.01 | 0.02 | 0.01 | 0.01 | 0.00 | 0.15 | 0.00 |
| $\gamma_{wave}$ | 0.00 | 0.00 | 0.00 | 0.00 | 0.00 | 0.00 | 0.00 | 0.00 | 0.00 | 0.00 | 0.00 | 0.00 |
| $V_{current}$ | 0.02 | 0.02 | 0.02 | 0.02 | 0.00 | 0.02 | 0.03 | 0.42 | 0.40 | 0.51 | 0.01 | 0.00 |
| $\Theta_{current}$ | 0.17 | 0.09 | 0.16 | 0.14 | 0.27 | 0.27 | 0.14 | 0.26 | 0.27 | 0.51 | 0.26 | 0.08 |
| $Cd_{upper}$ | 0.00 | 0.02 | 0.00 | 0.00 | 0.00 | 0.00 | 0.00 | 0.00 | 0.00 | 0.00 | 0.00 | 0.00 |
| $Cd_{lower}$ | 0.00 | 0.00 | 0.00 | 0.00 | 0.00 | 0.00 | 0.00 | 0.14 | 0.14 | 0.01 | 0.00 | 0.00 |
| $Cd_{axial}$ | 0.00 | 0.00 | 0.00 | 0.00 | 0.00 | 0.00 | 0.00 | 0.00 | 0.00 | 0.00 | 0.00 | 0.00 |
| $Cd_{rectangular}$ | 0.00 | 0.02 | 0.00 | 0.00 | 0.02 | 0.00 | 0.00 | 0.00 | 0.00 | 0.00 | 0.00 | 0.00 |

**Table B1.** Fraction of all perturbations of a given input parameter that exceed the significant ultimate EE threshold for a given QOI for the IEA 15 MW RWT on the VolturnUS-S platform in the Gulf of Maine





| | Root Bending Mom. | Root Pitching Mom. | LS Shaft Bending Mom. | Yaw-bearing Bending Mom. | Yaw-bearing Yaw Mom. | Twr. Base Mom. | Blade Tip Deflection | Fairlead Tension | Anchor Tension | Watch Circle | Heel Angle | Nacelle Acceleration |
|---|---|---|---|---|---|---|---|---|---|---|---|---|
| $U_{mean}$ | 0.05 | 0.06 | 0.05 | 0.05 | 0.05 | 0.05 | 0.05 | 0.00 | 0.00 | 0.01 | 0.00 | 0.03 |
| $Shear$ | 0.00 | 0.00 | 0.00 | 0.00 | 0.00 | 0.00 | 0.00 | 0.00 | 0.00 | 0.00 | 0.00 | 0.00 |
| $Veer$ | 0.00 | 0.00 | 0.00 | 0.00 | 0.00 | 0.00 | 0.00 | 0.00 | 0.00 | 0.00 | 0.00 | 0.00 |
| $\gamma_{wind}$ | 0.00 | 0.00 | 0.00 | 0.00 | 0.02 | 0.00 | 0.00 | 0.00 | 0.00 | 0.00 | 0.00 | 0.00 |
| $\sigma_u$ | 0.00 | 0.00 | 0.00 | 0.00 | 0.00 | 0.01 | 0.00 | 0.00 | 0.00 | 0.00 | 0.00 | 0.00 |
| $L_u$ | 0.00 | 0.00 | 0.00 | 0.00 | 0.00 | 0.01 | 0.00 | 0.00 | 0.00 | 0.00 | 0.00 | 0.00 |
| $a_u$ | 0.00 | 0.02 | 0.00 | 0.00 | 0.02 | 0.00 | 0.00 | 0.00 | 0.00 | 0.00 | 0.00 | 0.00 |
| $b_u$ | 0.00 | 0.02 | 0.00 | 0.00 | 0.03 | 0.02 | 0.00 | 0.00 | 0.00 | 0.00 | 0.00 | 0.02 |
| $Cl_{tip}$ | 0.09 | 0.09 | 0.05 | 0.06 | 0.08 | 0.01 | 0.08 | 0.00 | 0.00 | 0.00 | 0.01 | 0.02 |
| $Cd_{tip}$ | 0.08 | 0.06 | 0.12 | 0.11 | 0.08 | 0.04 | 0.08 | 0.00 | 0.00 | 0.00 | 0.00 | 0.08 |
| $Cl_{root}$ | 0.02 | 0.02 | 0.02 | 0.02 | 0.02 | 0.02 | 0.02 | 0.00 | 0.00 | 0.00 | 0.01 | 0.00 |
| $Cd_{root}$ | 0.02 | 0.02 | 0.00 | 0.02 | 0.02 | 0.01 | 0.02 | 0.00 | 0.00 | 0.00 | 0.00 | 0.02 |
| $Cd_{Tower}$ | 0.02 | 0.02 | 0.00 | 0.00 | 0.02 | 0.01 | 0.02 | 0.00 | 0.00 | 0.00 | 0.00 | 0.05 |
| $Yaw$ | 0.14 | 0.14 | 0.16 | 0.16 | 0.23 | 0.22 | 0.12 | 0.00 | 0.00 | 0.00 | 0.02 | 0.17 |
| $Twist_{root}$ | 0.03 | 0.02 | 0.00 | 0.00 | 0.02 | 0.01 | 0.02 | 0.00 | 0.00 | 0.00 | 0.00 | 0.02 |
| $Twist_{tip}$ | 0.02 | 0.00 | 0.00 | 0.00 | 0.02 | 0.02 | 0.00 | 0.00 | 0.00 | 0.00 | 0.00 | 0.05 |
| $TK$ | 0.00 | 0.00 | 0.00 | 0.00 | 0.00 | 0.01 | 0.00 | 0.00 | 0.00 | 0.00 | 0.00 | 0.03 |
| $TD$ | 0.00 | 0.00 | 0.00 | 0.00 | 0.02 | 0.02 | 0.00 | 0.00 | 0.00 | 0.00 | 0.00 | 0.02 |
| $COG_X$ | 0.11 | 0.11 | 0.11 | 0.11 | 0.06 | 0.23 | 0.11 | 0.00 | 0.00 | 0.01 | 0.64 | 0.08 |
| $COG_Z$ | 0.00 | 0.00 | 0.00 | 0.00 | 0.00 | 0.01 | 0.00 | 0.00 | 0.00 | 0.00 | 0.00 | 0.00 |
| $IYY$ | 0.00 | 0.00 | 0.00 | 0.00 | 0.00 | 0.00 | 0.00 | 0.00 | 0.00 | 0.00 | 0.00 | 0.00 |
| $IZZ$ | 0.00 | 0.00 | 0.00 | 0.00 | 0.00 | 0.00 | 0.00 | 0.00 | 0.00 | 0.00 | 0.00 | 0.00 |
| $Mass$ | 0.00 | 0.00 | 0.00 | 0.00 | 0.00 | 0.00 | 0.00 | 0.00 | 0.00 | 0.00 | 0.00 | 0.02 |
| $K_{fiber}$ | 0.00 | 0.00 | 0.00 | 0.00 | 0.00 | 0.00 | 0.00 | 0.00 | 0.00 | 0.00 | 0.00 | 0.00 |
| $L_{fiber}$ | 0.14 | 0.14 | 0.17 | 0.16 | 0.12 | 0.10 | 0.14 | 0.62 | 0.62 | 0.51 | 0.06 | 0.09 |
| $M_{chain}$ | 0.00 | 0.00 | 0.00 | 0.00 | 0.00 | 0.00 | 0.00 | 0.00 | 0.00 | 0.00 | 0.00 | 0.00 |
| $Depth$ | 0.02 | 0.02 | 0.02 | 0.02 | 0.00 | 0.01 | 0.02 | 0.00 | 0.00 | 0.00 | 0.00 | 0.02 |
| $\Theta_{spread}$ | 0.00 | 0.00 | 0.00 | 0.00 | 0.00 | 0.00 | 0.00 | 0.00 | 0.00 | 0.00 | 0.00 | 0.00 |
| $\Theta_{mis}$ | 0.05 | 0.05 | 0.06 | 0.05 | 0.06 | 0.30 | 0.05 | 0.15 | 0.15 | 0.08 | 0.42 | 0.06 |
| $Hs$ | 0.00 | 0.00 | 0.00 | 0.00 | 0.00 | 0.00 | 0.00 | 0.00 | 0.00 | 0.00 | 0.03 | 0.00 |
| $H_{max}$ | 0.00 | 0.00 | 0.00 | 0.00 | 0.00 | 0.00 | 0.00 | 0.00 | 0.00 | 0.00 | 0.00 | 0.00 |
| $XH_{max}$ | 0.00 | 0.00 | 0.00 | 0.00 | 0.00 | 0.02 | 0.00 | 0.00 | 0.00 | 0.00 | 0.00 | 0.00 |
| $Tp$ | 0.02 | 0.00 | 0.00 | 0.00 | 0.02 | 0.02 | 0.02 | 0.00 | 0.00 | 0.00 | 0.12 | 0.02 |
| $\gamma_{wave}$ | 0.00 | 0.00 | 0.00 | 0.00 | 0.00 | 0.01 | 0.00 | 0.00 | 0.00 | 0.00 | 0.00 | 0.00 |
| $V_{current}$ | 0.00 | 0.00 | 0.00 | 0.00 | 0.00 | 0.00 | 0.00 | 0.00 | 0.00 | 0.00 | 0.00 | 0.00 |
| $\Theta_{current}$ | 0.16 | 0.16 | 0.19 | 0.17 | 0.19 | 0.19 | 0.16 | 0.24 | 0.24 | 0.43 | 0.10 | 0.17 |
| $Cd_{upper}$ | 0.00 | 0.00 | 0.00 | 0.00 | 0.00 | 0.00 | 0.00 | 0.00 | 0.00 | 0.00 | 0.00 | 0.00 |
| $Cd_{lower}$ | 0.00 | 0.02 | 0.02 | 0.02 | 0.02 | 0.01 | 0.00 | 0.00 | 0.00 | 0.01 | 0.00 | 0.00 |
| $Cd_{axial}$ | 0.00 | 0.00 | 0.00 | 0.00 | 0.00 | 0.00 | 0.00 | 0.00 | 0.00 | 0.00 | 0.00 | 0.00 |
| $Cd_{rectangular}$ | 0.00 | 0.00 | 0.00 | 0.00 | 0.00 | 0.00 | 0.00 | 0.00 | 0.00 | 0.00 | 0.00 | 0.00 |

**Table B2.** Fraction of all perturbations of a given input parameter that exceed the significant ultimate EE threshold for a given QOI for the IEA 15 MW RWT on the VolturnUS-S platform in Humboldt Bay





| | Root Bending Mom. | Root Pitching Mom. | LS Shaft Bending Mom. | Yaw-bearing Bending Mom. | Yaw-bearing Yaw Mom. | Twr. Base Mom. | Blade Tip Deflection | Fairlead Tension | Anchor Tension | Watch Circle | Heel Angle | Nacelle Acceleration |
|---|---|---|---|---|---|---|---|---|---|---|---|---|
| $U_{mean}$ | 0.00 | 0.00 | 0.06 | 0.00 | 0.00 | 0.00 | 0.00 | 0.00 | 0.00 | 0.00 | 0.00 | 0.00 |
| $Shear$ | 0.09 | 0.12 | 0.09 | 0.00 | 0.00 | 0.00 | 0.09 | 0.00 | 0.00 | 0.00 | 0.00 | 0.00 |
| $Veer$ | 0.22 | 0.25 | 0.06 | 0.00 | 0.00 | 0.00 | 0.09 | 0.00 | 0.00 | 0.03 | 0.03 | 0.00 |
| $\gamma_{wind}$ | 0.00 | 0.03 | 0.06 | 0.00 | 0.03 | 0.00 | 0.06 | 0.00 | 0.00 | 0.00 | 0.00 | 0.00 |
| $\sigma_u$ | 0.00 | 0.00 | 0.03 | 0.00 | 0.00 | 0.00 | 0.00 | 0.00 | 0.00 | 0.00 | 0.00 | 0.00 |
| $L_u$ | 0.00 | 0.00 | 0.00 | 0.00 | 0.00 | 0.00 | 0.00 | 0.00 | 0.00 | 0.00 | 0.03 | 0.00 |
| $a_u$ | 0.00 | 0.03 | 0.03 | 0.00 | 0.00 | 0.00 | 0.03 | 0.03 | 0.00 | 0.00 | 0.03 | 0.00 |
| $b_u$ | 0.00 | 0.03 | 0.03 | 0.00 | 0.00 | 0.00 | 0.03 | 0.00 | 0.00 | 0.00 | 0.06 | 0.00 |
| $Cl_{tip}$ | 0.06 | 0.09 | 0.06 | 0.00 | 0.00 | 0.00 | 0.06 | 0.00 | 0.00 | 0.00 | 0.00 | 0.00 |
| $Cd_{tip}$ | 0.00 | 0.00 | 0.06 | 0.00 | 0.00 | 0.00 | 0.06 | 0.00 | 0.00 | 0.00 | 0.00 | 0.00 |
| $Cl_{root}$ | 0.03 | 0.00 | 0.00 | 0.00 | 0.00 | 0.00 | 0.06 | 0.00 | 0.00 | 0.00 | 0.00 | 0.00 |
| $Cd_{root}$ | 0.00 | 0.00 | 0.00 | 0.00 | 0.00 | 0.00 | 0.00 | 0.00 | 0.00 | 0.00 | 0.00 | 0.00 |
| $Cd_{Tower}$ | 0.00 | 0.00 | 0.00 | 0.00 | 0.00 | 0.00 | 0.00 | 0.00 | 0.00 | 0.00 | 0.00 | 0.00 |
| $Yaw$ | 0.09 | 0.22 | 0.12 | 0.00 | 0.00 | 0.00 | 0.12 | 0.03 | 0.00 | 0.03 | 0.00 | 0.03 |
| $Twist_{root}$ | 0.28 | 0.19 | 0.09 | 0.00 | 0.00 | 0.00 | 0.06 | 0.00 | 0.00 | 0.12 | 0.03 | 0.00 |
| $Twist_{tip}$ | 0.16 | 0.22 | 0.06 | 0.00 | 0.03 | 0.00 | 0.06 | 0.00 | 0.00 | 0.06 | 0.03 | 0.00 |
| $TK$ | 0.00 | 0.00 | 0.00 | 0.00 | 0.00 | 0.00 | 0.00 | 0.00 | 0.00 | 0.00 | 0.00 | 0.00 |
| $TD$ | 0.00 | 0.00 | 0.00 | 0.00 | 0.00 | 0.00 | 0.00 | 0.00 | 0.00 | 0.00 | 0.00 | 0.00 |
| $COG_X$ | 0.03 | 0.09 | 0.06 | 0.00 | 0.00 | 0.00 | 0.06 | 0.00 | 0.00 | 0.00 | 0.47 | 0.00 |
| $COG_Z$ | 0.00 | 0.00 | 0.00 | 0.00 | 0.00 | 0.00 | 0.00 | 0.00 | 0.00 | 0.00 | 0.00 | 0.00 |
| $IYY$ | 0.00 | 0.00 | 0.00 | 0.00 | 0.00 | 0.00 | 0.00 | 0.00 | 0.00 | 0.00 | 0.00 | 0.00 |
| $IZZ$ | 0.00 | 0.00 | 0.00 | 0.00 | 0.00 | 0.00 | 0.00 | 0.00 | 0.00 | 0.00 | 0.00 | 0.00 |
| $Mass$ | 0.00 | 0.00 | 0.00 | 0.00 | 0.00 | 0.00 | 0.00 | 0.00 | 0.00 | 0.00 | 0.00 | 0.00 |
| $K_{fiber}$ | 0.00 | 0.00 | 0.00 | 0.00 | 0.00 | 0.00 | 0.00 | 0.00 | 0.00 | 0.00 | 0.00 | 0.00 |
| $L_{fiber}$ | 0.00 | 0.06 | 0.06 | 0.00 | 0.00 | 0.00 | 0.06 | 0.00 | 0.00 | 0.03 | 0.00 | 0.00 |
| $M_{chain}$ | 0.00 | 0.00 | 0.00 | 0.00 | 0.00 | 0.00 | 0.00 | 0.00 | 0.00 | 0.00 | 0.00 | 0.00 |
| $Depth$ | 0.00 | 0.03 | 0.03 | 0.00 | 0.00 | 0.00 | 0.03 | 0.00 | 0.00 | 0.00 | 0.00 | 0.00 |
| $\Theta_{spread}$ | 0.00 | 0.00 | 0.00 | 0.00 | 0.00 | 0.00 | 0.00 | 0.00 | 0.00 | 0.00 | 0.00 | 0.00 |
| $\Theta_{mis}$ | 0.16 | 0.06 | 0.09 | 0.50 | 0.66 | 0.44 | 0.06 | 0.25 | 0.28 | 0.25 | 0.34 | 0.44 |
| $Hs$ | 0.00 | 0.06 | 0.06 | 0.25 | 0.28 | 0.59 | 0.06 | 0.59 | 0.66 | 0.56 | 0.38 | 0.62 |
| $H_{max}$ | 0.00 | 0.00 | 0.00 | 0.00 | 0.00 | 0.00 | 0.00 | 0.00 | 0.00 | 0.00 | 0.00 | 0.00 |
| $XH_{max}$ | 0.00 | 0.00 | 0.03 | 0.00 | 0.00 | 0.00 | 0.03 | 0.00 | 0.00 | 0.00 | 0.00 | 0.00 |
| $Tp$ | 0.00 | 0.00 | 0.03 | 0.31 | 0.44 | 0.56 | 0.00 | 0.22 | 0.28 | 0.22 | 0.03 | 0.34 |
| $\gamma_{wave}$ | 0.00 | 0.00 | 0.03 | 0.00 | 0.00 | 0.00 | 0.00 | 0.00 | 0.00 | 0.00 | 0.00 | 0.00 |
| $V_{current}$ | 0.03 | 0.03 | 0.06 | 0.00 | 0.00 | 0.00 | 0.06 | 0.09 | 0.09 | 0.09 | 0.00 | 0.00 |
| $\Theta_{current}$ | 0.00 | 0.00 | 0.00 | 0.00 | 0.00 | 0.00 | 0.06 | 0.19 | 0.09 | 0.12 | 0.03 | 0.00 |
| $Cd_{upper}$ | 0.00 | 0.00 | 0.03 | 0.00 | 0.00 | 0.00 | 0.00 | 0.00 | 0.00 | 0.00 | 0.00 | 0.00 |
| $Cd_{lower}$ | 0.00 | 0.00 | 0.00 | 0.00 | 0.00 | 0.00 | 0.03 | 0.00 | 0.00 | 0.00 | 0.00 | 0.00 |
| $Cd_{axial}$ | 0.00 | 0.00 | 0.00 | 0.00 | 0.00 | 0.00 | 0.00 | 0.00 | 0.00 | 0.00 | 0.00 | 0.00 |
| $Cd_{rectangular}$ | 0.00 | 0.00 | 0.00 | 0.00 | 0.00 | 0.00 | 0.00 | 0.00 | 0.00 | 0.00 | 0.00 | 0.00 |

**Table B3.** Fraction of all perturbations of a given input parameter that exceed the significant fatigue EE threshold for a given QOI for the IEA 15 MW RWT on the VolturnUS-S platform in the Gulf of Maine





| | Root Bending Mom. | Root Pitching Mom. | LS Shaft Bending Mom. | Yaw-bearing Bending Mom. | Yaw-bearing Yaw Mom. | Twr. Base Mom. | Blade Tip Deflection | Fairlead Tension | Anchor Tension | Watch Circle | Heel Angle | Nacelle Acceleration |
|---|---|---|---|---|---|---|---|---|---|---|---|---|
| $U_{mean}$ | 0.00 | 0.00 | 0.03 | 0.00 | 0.00 | 0.00 | 0.00 | 0.00 | 0.00 | 0.00 | 0.00 | 0.00 |
| $Shear$ | 0.00 | 0.03 | 0.00 | 0.00 | 0.00 | 0.00 | 0.06 | 0.00 | 0.00 | 0.00 | 0.00 | 0.00 |
| $Veer$ | 0.06 | 0.03 | 0.06 | 0.00 | 0.00 | 0.00 | 0.00 | 0.00 | 0.00 | 0.03 | 0.00 | 0.00 |
| $\gamma_{wind}$ | 0.00 | 0.03 | 0.00 | 0.00 | 0.16 | 0.00 | 0.06 | 0.00 | 0.00 | 0.00 | 0.00 | 0.00 |
| $\sigma_u$ | 0.00 | 0.03 | 0.00 | 0.00 | 0.09 | 0.00 | 0.00 | 0.00 | 0.00 | 0.00 | 0.00 | 0.00 |
| $L_u$ | 0.00 | 0.00 | 0.00 | 0.00 | 0.00 | 0.00 | 0.00 | 0.03 | 0.03 | 0.00 | 0.03 | 0.00 |
| $a_u$ | 0.00 | 0.00 | 0.00 | 0.00 | 0.00 | 0.00 | 0.00 | 0.00 | 0.00 | 0.00 | 0.09 | 0.00 |
| $b_u$ | 0.03 | 0.03 | 0.00 | 0.00 | 0.00 | 0.00 | 0.00 | 0.00 | 0.00 | 0.00 | 0.09 | 0.00 |
| $Cl_{tip}$ | 0.03 | 0.03 | 0.03 | 0.00 | 0.03 | 0.00 | 0.06 | 0.00 | 0.00 | 0.00 | 0.00 | 0.03 |
| $Cd_{tip}$ | 0.00 | 0.03 | 0.06 | 0.00 | 0.00 | 0.00 | 0.06 | 0.00 | 0.00 | 0.00 | 0.00 | 0.00 |
| $Cl_{root}$ | 0.00 | 0.03 | 0.03 | 0.00 | 0.00 | 0.00 | 0.03 | 0.00 | 0.00 | 0.03 | 0.00 | 0.00 |
| $Cd_{root}$ | 0.00 | 0.00 | 0.00 | 0.00 | 0.00 | 0.00 | 0.00 | 0.00 | 0.00 | 0.00 | 0.00 | 0.00 |
| $Cd_{Tower}$ | 0.00 | 0.00 | 0.00 | 0.00 | 0.00 | 0.00 | 0.00 | 0.00 | 0.00 | 0.00 | 0.00 | 0.00 |
| $Yaw$ | 0.09 | 0.16 | 0.09 | 0.03 | 0.03 | 0.00 | 0.12 | 0.03 | 0.03 | 0.09 | 0.03 | 0.03 |
| $Twist_{root}$ | 0.06 | 0.06 | 0.03 | 0.00 | 0.03 | 0.00 | 0.06 | 0.00 | 0.00 | 0.06 | 0.00 | 0.00 |
| $Twist_{tip}$ | 0.09 | 0.03 | 0.00 | 0.00 | 0.03 | 0.00 | 0.00 | 0.00 | 0.00 | 0.06 | 0.00 | 0.00 |
| $TK$ | 0.00 | 0.00 | 0.00 | 0.00 | 0.00 | 0.00 | 0.00 | 0.00 | 0.00 | 0.00 | 0.00 | 0.00 |
| $TD$ | 0.00 | 0.00 | 0.00 | 0.00 | 0.00 | 0.00 | 0.00 | 0.00 | 0.00 | 0.00 | 0.00 | 0.00 |
| $COG_X$ | 0.06 | 0.12 | 0.06 | 0.03 | 0.09 | 0.00 | 0.09 | 0.00 | 0.00 | 0.03 | 0.44 | 0.03 |
| $COG_Z$ | 0.00 | 0.00 | 0.00 | 0.00 | 0.00 | 0.00 | 0.00 | 0.00 | 0.00 | 0.00 | 0.00 | 0.00 |
| $IYY$ | 0.00 | 0.00 | 0.00 | 0.00 | 0.00 | 0.00 | 0.00 | 0.00 | 0.00 | 0.00 | 0.00 | 0.00 |
| $IZZ$ | 0.00 | 0.00 | 0.00 | 0.00 | 0.00 | 0.00 | 0.00 | 0.00 | 0.00 | 0.00 | 0.00 | 0.00 |
| $Mass$ | 0.00 | 0.00 | 0.00 | 0.00 | 0.00 | 0.00 | 0.00 | 0.00 | 0.00 | 0.00 | 0.00 | 0.00 |
| $K_{fiber}$ | 0.00 | 0.00 | 0.00 | 0.00 | 0.00 | 0.00 | 0.00 | 0.00 | 0.00 | 0.00 | 0.00 | 0.00 |
| $L_{fiber}$ | 0.12 | 0.12 | 0.09 | 0.09 | 0.03 | 0.03 | 0.12 | 0.31 | 0.31 | 0.22 | 0.00 | 0.03 |
| $M_{chain}$ | 0.00 | 0.00 | 0.00 | 0.00 | 0.00 | 0.00 | 0.00 | 0.00 | 0.00 | 0.00 | 0.00 | 0.00 |
| $Depth$ | 0.00 | 0.00 | 0.00 | 0.00 | 0.00 | 0.00 | 0.00 | 0.00 | 0.00 | 0.00 | 0.00 | 0.00 |
| $\Theta_{spread}$ | 0.00 | 0.00 | 0.00 | 0.00 | 0.00 | 0.00 | 0.00 | 0.00 | 0.00 | 0.00 | 0.00 | 0.00 |
| $\Theta_{mis}$ | 0.28 | 0.12 | 0.06 | 0.69 | 0.44 | 0.38 | 0.06 | 0.25 | 0.22 | 0.09 | 0.50 | 0.31 |
| $Hs$ | 0.03 | 0.00 | 0.03 | 0.03 | 0.06 | 0.12 | 0.00 | 0.53 | 0.50 | 0.03 | 0.06 | 0.03 |
| $H_{max}$ | 0.00 | 0.00 | 0.00 | 0.00 | 0.00 | 0.00 | 0.00 | 0.00 | 0.00 | 0.00 | 0.00 | 0.00 |
| $XH_{max}$ | 0.00 | 0.00 | 0.00 | 0.00 | 0.00 | 0.00 | 0.00 | 0.00 | 0.00 | 0.00 | 0.00 | 0.00 |
| $Tp$ | 0.06 | 0.00 | 0.00 | 0.25 | 0.25 | 0.44 | 0.00 | 0.53 | 0.53 | 0.12 | 0.19 | 0.38 |
| $\gamma_{wave}$ | 0.00 | 0.00 | 0.00 | 0.00 | 0.00 | 0.00 | 0.00 | 0.00 | 0.00 | 0.00 | 0.00 | 0.00 |
| $V_{current}$ | 0.00 | 0.00 | 0.00 | 0.00 | 0.00 | 0.00 | 0.00 | 0.00 | 0.00 | 0.06 | 0.00 | 0.00 |
| $\Theta_{current}$ | 0.03 | 0.03 | 0.03 | 0.06 | 0.03 | 0.03 | 0.03 | 0.06 | 0.06 | 0.09 | 0.00 | 0.03 |
| $Cd_{upper}$ | 0.00 | 0.00 | 0.00 | 0.00 | 0.00 | 0.00 | 0.00 | 0.00 | 0.00 | 0.00 | 0.00 | 0.00 |
| $Cd_{lower}$ | 0.00 | 0.00 | 0.00 | 0.00 | 0.00 | 0.00 | 0.00 | 0.00 | 0.00 | 0.03 | 0.00 | 0.00 |
| $Cd_{axial}$ | 0.00 | 0.00 | 0.00 | 0.00 | 0.00 | 0.00 | 0.00 | 0.00 | 0.00 | 0.00 | 0.00 | 0.00 |
| $Cd_{rectangular}$ | 0.00 | 0.00 | 0.00 | 0.00 | 0.00 | 0.00 | 0.00 | 0.00 | 0.00 | 0.00 | 0.00 | 0.00 |

**Table B4.** Fraction of all perturbations of a given input parameter that exceed the significant fatigue EE threshold for a given QOI for the IEA 15 MW RWT on the VolturnUS-S platform in Humboldt Bay



## Appendix C:  Fatigue Calculation Comparison

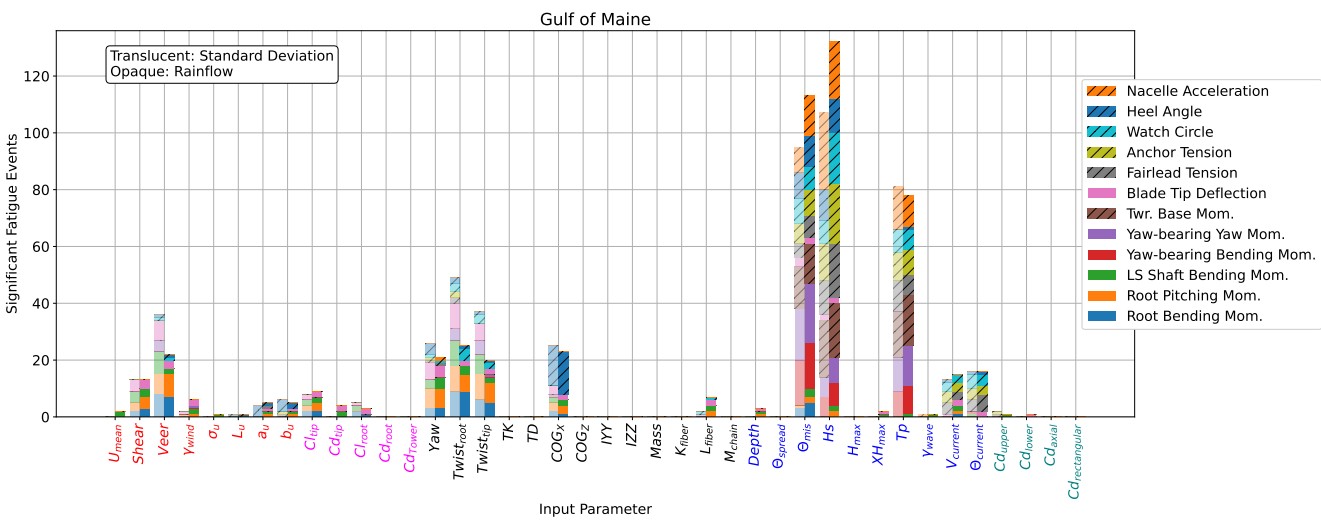

**Figure C1.** Sensitivity of fatigue loads EE values for the IEA 15 MW RWT on the VolturnUS-S platform in the Gulf of Maine when calculated with a rainflow counter and Miner's rule compared to the standard deviation

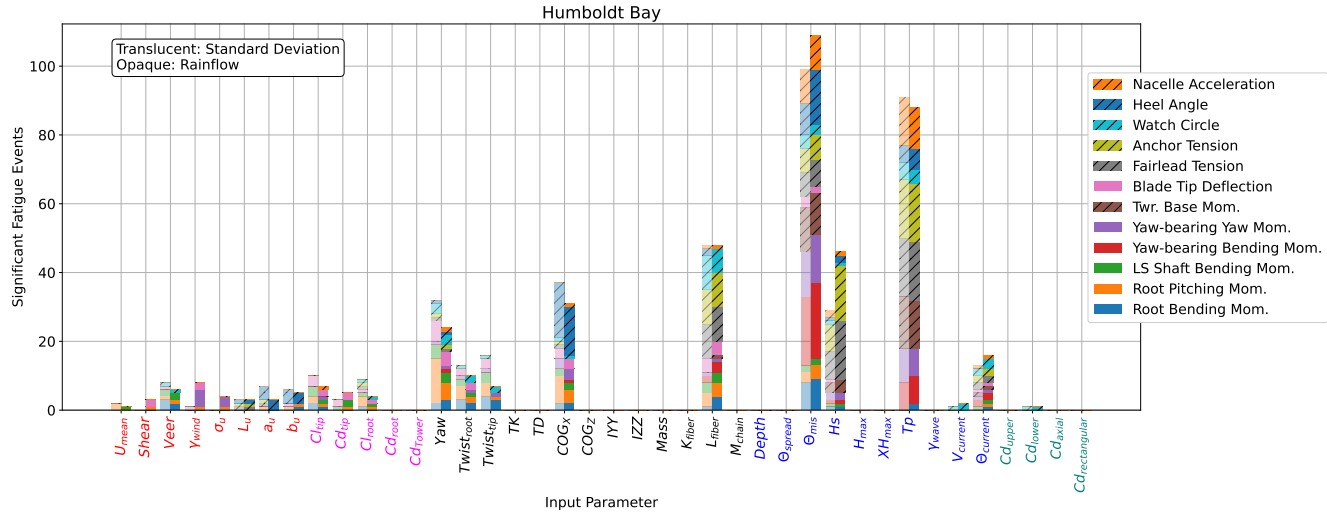

**Figure C2.** Sensitivity of fatigue loads EE values for the IEA 15 MW RWT on the VolturnUS-S platform in Humboldt Bay when calculated with a rainflow counter and Miner's rule compared to the standard deviation



## Appendix D:  Additional Checks of Seed Convergence

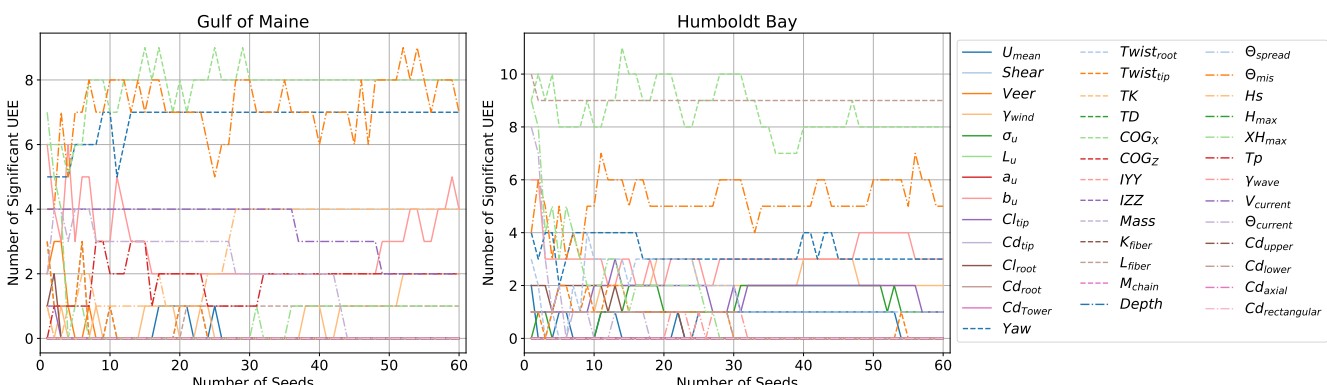

**Figure D1.** Convergence of ultimate load EE sensitivity for the IEA 15 MW RWT on the VolturnUS-S platform with an increasing number of random seed numbers for one starting point only based on the nominal values for each input parameter

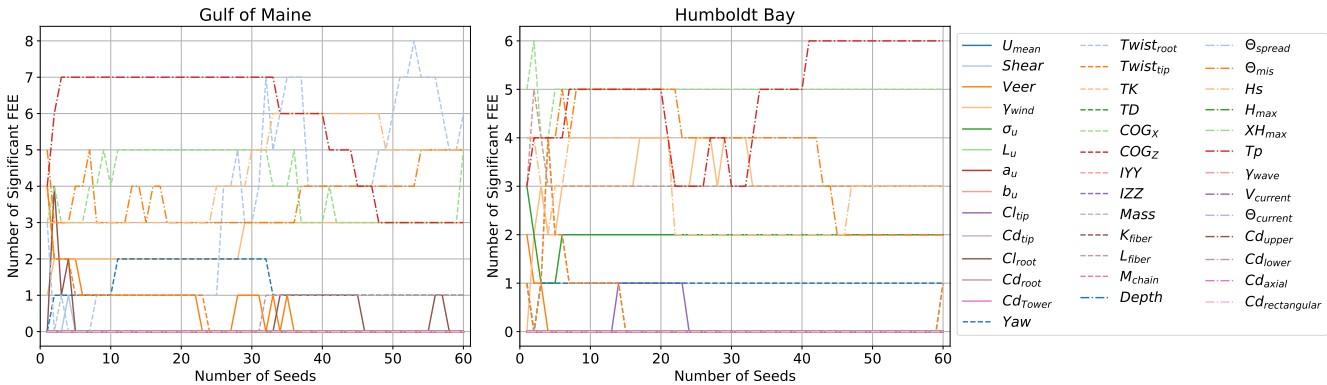

**Figure D2.** Convergence of fatigue load EE sensitivity for the IEA 15 MW RWT on the VolturnUS-S platform with an increasing number of random seed numbers for one starting point only based on the nominal values for each input parameter




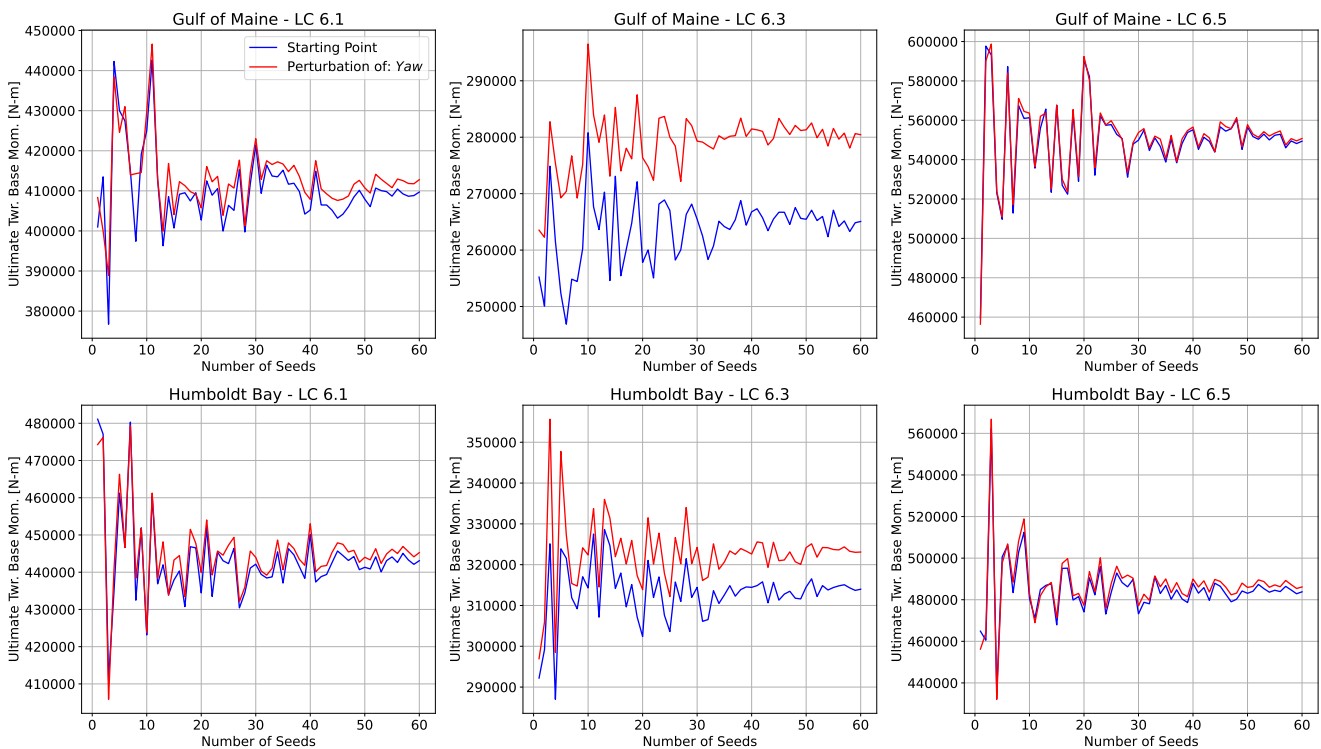

**Figure D3.** Seed convergence of the ultimate root pitching moment due to a change in the nacelle yaw (blue line: nominal starting point, red line: perturbation in yaw)

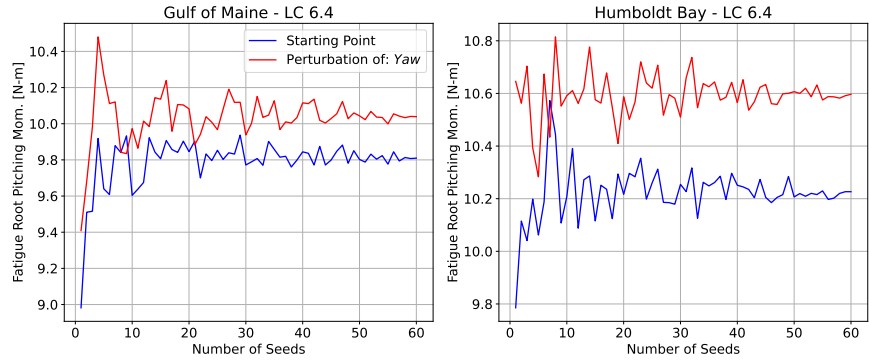

**Figure D4.** Seed convergence of the fatigue root pitching moment due to a change in the nacelle yaw (blue line: nominal starting point, red line: perturbation in yaw)





*Author contributions.* The project was conceptualized by JJ and AR, who provided coordination and direction of all aspects. The input parameter data processing and numerical model execution was performed by WW. The draft text was prepared by WW with editing and review from JJ and AR.

*Competing interests.* At least one of the (co-)authors is a member of the editorial board of *Wind Energy Science*. The authors have no other competing interests to declare.

*Disclaimer.* This work was authored by the National Renewable Energy Laboratory, operated by the Alliance for Sustainable Energy, LLC, for the U.S. Department of Energy (DOE) under contract no. DE-AC36-08GO28308. Funding provided by U.S. Department of Energy Office of Energy Efficiency and Renewable Energy Wind Energy Technologies Office. The views expressed in the article do not necessarily represent the views of the DOE or the U.S. Government. The U.S. Government retains and the publisher, by accepting the article for publication, acknowledges that the U.S. Government retains a nonexclusive, paid-up, irrevocable, worldwide license to publish or reproduce the published form of this work, or allow others to do so, for U.S. Government purposes.

*Acknowledgements.* We would like to thank Lu Wang of NREL for performing the potential flow calculations for the VolturnUS-S platform, and Ericka Lozon et al. of NREL for providing the site-specific mooring systems used in this analysis (Lozon et al., 2024).

A portion of the research was performed using computational resources sponsored by the Department of Energy's Office of Energy Efficiency and Renewable Energy and located at NREL.





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
