# Peer review of "Sensitivity analysis of numerical modeling input parameters on floating offshore wind turbine loads in extreme idling conditions"

_Wind Energy Science, 2024_

## Referee Comment (RC1)

Review of WES-2024-130

**Sensitivity analysis of numerical modeling input parameters on floating offshore wind turbine loads in extreme idling conditions**

The study proposes a global sensitivity analysis to model input parameters to floating wind turbine numerical models. The study analyses the effect on extreme and fatigue loads in a selection of non-operational IEC load cases of more than 50 input parameters. The study is well written and the topic is very relevant. Such an analysis can help quantify uncertainties in numerical input parameter. The outcomes can help inform numerical modelers and also technical experts. The study is quite intricate and various aspects of it are complex. Some of the suggestions given below are directly linked to such complexity. Some remarks and suggestions to improve the draft submission are given below:

L98: IEC load cases do not typically consider Veer. Coul the authors comment on why they considered it in this study?

L107-L117: Distinguishing between physical and numerical instabilities in idling simulations can be challenging for numerical modelers. Increasing damping ratio as suggested in this study may sound somewhat arbitrary. It would be of great value to add any suggestions or experience-driven guidance on this. Connected to this, why was this parameter not considered in the sensitivity analysis as tower damping was?

L133: is there a reason for choosing 10%? Was this based on preliminary analyses? The question arises from the fact that 10% is arguably not that small of a variation

L137: So each simulation uses a different wind/wave seed and the number of simulations is high enough to ensure statistical convergence in each test point?

L153: It is important to weigh variations that are more likely to generate ultimate loading more, and adding the ultimate load to the variation in QOI is a way of doing this. However, this manuscript focuses on sensitivity to input parameters in uncertain engineering models. In principle, the extreme load that is added to the variation could also be uncertain, and somewhat skew the results of his analysis. Did authors observe relevant differences in the conclusions if the ultimate load for each DLC is not added to the variation in QOI?

L172: What do you mean for "The coefficients at the mid-span sections between the tip and the root use a coefficient modification that is linearly interpolated between the tip and root."? The coefficients of mid-span airfoils are modified from their respective polars proportionally to their distance from tip/root? Please clarify in the text

Fig. 7: This figure could be commented in more depth. What is the cause of the bi-modal distribution in Humboldt Bay? Swell? In Gulf of Maine the waves also do not appear to be wind driven as one would expect wind-wave misalignment close to zero at high wind speeds (LC 6.1-6.5)

L273, Fig. 9 – IEC recommends the IFORM method to derive environmental contours, how does this differ to the principal component analysis performed here? The entire paragraph L273-L282 could benefit from some additional clarity as it is not clear for me what underlying complexity the authors are trying to solve here.

This may sound surprising, but I am somewhat uncertain about what is being shown in Figures 11-14. If I understand correctly Eq. 3 is used to compute the EE value for each parameter variation in each DLC. For each parameter multiple variations around multiple "mean" values are imposed. What is the rationale for post-processing this data? Only the most relevant variations need to be extracted? I would suggest expanding

L316-318 as I find this part quite intricate for a fist time reader. If possible, an equation to reference when computing the "significant Ultimate Events" would go a long way here.

L337: Possibly due to the uncertainty highlighted in my previous comment, what is the need for normalization?

L363-L382: If, as appears, the directionality of wind and waves matter, rather than the misalignment, then the absolute orientation of the platform also matters. This would make it quite hard to define "general" load cases, which has been the industry-standard approach up to this point. Perhaps a comment on this aspect could be added here or in the conclusions.

Conclusions: It is particularly interesting that directionality parameters seem to have a large influence on quantities which should compensate for this by considering the sum of X and Y moments, such as the total tower base bending moment. An explanation on why this may be would be nice to see in the conclusions.

---

## Author Comment (AC1)

Dear referees,

Thank you for your time providing thorough feedback and suggestions for our paper. Please find our answers to your comments and the corresponding updates to the paper below. We hope to have addressed all of your comments, improving the manuscript. We will be happy to have continued discussion. We will submit the revised manuscript including a pdf highlighting the differences made to the text in the next few days.

Thank you,

Will Wiley, Jason Jonkman, and Amy Robertson

**Referee 1**

Comments:

*The study proposes a global sensitivity analysis to model input parameters to floating wind turbine numerical models. The study analyses the effect on extreme and fatigue loads in a selection of non-operational IEC load cases of more than 50 input parameters. The study is well written and the topic is very relevant. Such an analysis can help quantify uncertainties in numerical input parameter. The outcomes can help inform numerical modelers and also technical experts. The study is quite intricate and various aspects of it are complex. Some of the suggestions given below are directly linked to such complexity. Some remarks and suggestions to improve the draft submission are given below:*
Thank you for the feedback.
N/A

*L98: IEC load cases do not typically consider veer. Coul the authors comment on why they considered it in this study?*
It was thought that with the minimal loading on the idling rotor, that any source of asymmetry or misalignment could cause important changes to the loads. Generally, incident wind parameters were found to have a low relative sensitivity in the above-cut-out load cases, but for the Gulf of Maine, veer did result in the highest number of significant ultimate load and fatigue load events out of all included incident wind parameters. Large eddy simulations of hurricanes have shown that high veer is likely to be present in extreme conditions (Sanchez Gomez et al. 2023 10.1029/2023JD039233). Additionally, as rotor sizes increase, the impact of veer will likely grow.
Addition to end of Section 3 Paragraph 2: "Although veer is not defined by IEC for the relevant load cases, it was included, as any source of asymmetry or imbalance for the idling rotor was expected to have potentially important changes to loads. Large eddy simulations of hurricanes have shown that high veer is likely to be present in extreme conditions (Sanchez Gomez et al., 2023). Additionally, as rotor sizes increase, the impact of veer will likely grow."

*L107-L117: Distinguishing between physical and numerical instabilities in idling simulations can be challenging for numerical modelers. Increasing damping ratio as suggested in this study may sound*

*somewhat arbitrary. It would be of great value to add any suggestions or experience-driven guidance on this. Connected to this, why was this parameter not considered in the sensitivity analysis as tower damping was?*

The authors agree with the challenges mentioned separating physical and numerical instability, and with the arbitrary nature of changing the damping ratio. Especially given the high uncertainty in blade damping ratio, this would be an interesting parameter to include in the analysis. Practically, it would be difficult to include; many values would result in a diverging simulations, not allowing the calculation of load values for comparison. Properly addressing this issue, e.g., by improving the aero-elastic damping in deep stall by changing the unsteady airfoil aerodynamic hysteresis loops, is important future work.

N/A

*L133: is there a reason for choosing 10%? Was this based on preliminary analyses? The question arises from the fact that 10% is arguably not that small of a variation*

The value of 10% has been used in past studies using the same EE sensitivity approach. The perturbations are 10% of the range of uncertainty, which may be small depending on the uncertainty, but is likely still larger than a traditional epsilon value for derivative calculations. If the perturbation is too small the effect on loads can be lost relative to the influence of seed number. This choice was thought to be "not too small to avoid numerical issues and not too large to smooth out nonlinearities" (Robertson et al., 2018).

Addition to the start of Section 2.4 Paragraph 2 "… 10% of the parameter range. This value has been used in previous EE analyses, and aims to result in significant changes in loads without smoothing out nonlinearities (Robertson et al., 2018)."

*L137: So each simulation uses a different wind/wave seed and the number of simulations is high enough to ensure statistical convergence in each test point?*

Each starting point in the parameter hyperspace was run with a set of different seed numbers. The perturbations from that starting point were run with the same set of seed numbers as the starting point. The number of starting points and the number of seed numbers was increased, and the resulting EE values were tracked as a function of how many seed numbers were used, to ensure that the differences between the quantities of interest in the perturbation and the starting point were independent of the seed number used. This convergence is shown in Fig. 18, 19, D1, D2, D3, and D4. It was determined that 20 different seed numbers were sufficient for each starting point, and this number was used in the analysis.

N/A

*L153: It is important to weigh variations that are more likely to generate ultimate loading more, and adding the ultimate load to the variation in QOI is a way of doing this. However, this manuscript focuses on sensitivity to input parameters in uncertain engineering models. In principle, the extreme load that is added to the variation could also be uncertain, and somewhat skew the results of his analysis. Did authors observe relevant differences in the conclusions if the ultimate load for each DLC is not added to the variation in QOI?*

The ultimate load EE sensitivities were not specifically compared without the addition of the nominal value. Figures A1 and A2 show strong load-case-based stratification of ultimate loads for some quantities of interest. For example, the ultimate fairlead and anchor tensions in the Gulf of Maine, shown in Figure A1, are significantly lower in DLC 6.3 compared to DLC 6.1 and DLC 6.5. To fairly assess sensitivity across the three ultimate load cases, this was accounted for by adding the nominal load value for the load case.

This is not necessary if the sensitivity for only a specific load case is of interest. Figures 13 and 14 show the ultimate load EE sensitivities for the load cases individually, where the nominal value does not make an impact. Differences in the relative sensitivity values between individual load cases and the aggregate, are influenced by the nominal load values.

N/A

*L172: What do you mean for "The coefficients at the mid-span sections between the tip and the root use a coefficient modification that is linearly interpolated between the tip and root."? The coefficients of mid-span airfoils are modified from their respective polars proportionally to their distance from tip/root? Please clarify in the text*

The text was reworded to more clearly explain the blade aerodynamic coefficients. The variable parameters define a change to the nominal coefficient for a given blade section. The factors are varied independently at the tip and root. The blade sections between the tip and root use a change factor that is linearly interpolated between the two parameters defined at the tip and root.

"The parameters denote a fractional change from the defined value for the airfoil. The coefficients between the tip and the root were modified with a factor linearly interpolated between the tip and root parameters."

*Fig. 7: This figure could be commented in more depth. What is the cause of the bi-modal distribution in Humboldt Bay? Swell? In Gulf of Maine the waves also do not appear to be wind driven as one would expect wind-wave misalignment close to zero at high wind speeds (LC 6.1-6.5)*

In the Humboldt Bay data, there is one dominant wave direction and two common wind directions. This results in the bi-modal distribution of misalignment for "All Data" in Figure 7. The Gulf of Maine data includes a large cluster of points with one wave direction and a full range of wind directions, and a large cluster of points where the wind and wave direction are roughly the same. The high probability of misalignment for wind speeds of the extreme events is interesting. Given the spread of the distributions, the selected misalignment ranges were at least 0.0 – 150.0 degrees for all load cases, covering close to all possibilities.

Added to the end of Section 3.1.3 Paragraph 3: "The distribution for the Humboldt Bay data is bimodal, due to a single dominant wave direction and two dominant wind directions. Even with the wind speed filtering, all load cases resulted in close to a full range of possible misalignment angles."

*L273, Fig. 9 – IEC recommends the IFORM method to derive environmental contours, how does this differ to the principal component analysis performed here? The entire paragraph L273-L282 could benefit from some additional clarity as it is not clear for me what underlying complexity the authors are trying to solve here.*

The method for determining the contours was based on work published by Sandia National Laboratory focused on predicting extreme sea states. The I-FORM method is still used, but the data is first treated with principal component analysis to "create an orthogonal decomposition of the data," which was shown to better represent measured data (Eckert-Gallup et al., 2016). The issue mentioned in the second sentence of the paragraph is that the recorded data in NDBC is a frequency spectrum, and the peak period is simply the inverse of the frequency with the highest energy, only allowing a set of values along the selected frequency bins. The mentioned smoothing function was a curve fit to result in a more realistic distribution of peak periods.

Addition to Section 3.1.3 Paragraph 6: "This approach includes additional data processing prior to implementing the standard I-FORM method, which has been shown to better represent measured data of extreme events (Eckert-Gallup et al., 2016)."

*This may sound surprising, but I am somewhat uncertain about what is being shown in Figures 11-14. If I understand correctly Eq. 3 is used to compute the EE value for each parameter variation in each DLC. For each parameter multiple variations around multiple "mean" values are imposed. What is the rationale for post-processing this data? Only the most relevant variations need to be extracted? I would suggest expanding*

Figure 3 shows a simplified three-parameter visual of the sampling methodology. Each starting point shown in blue, represents some combination of parameters in the hyperspace. The red points extending from each blue point then show some small change in only one parameter from the blue starting point. The comparison of loads between the perturbation and the starting point result in a kind of local partial derivative. The number of starting points is increased to assess the sensitivity across a full range of possible parameter combinations. Equation 3 shows the calculation of this local partial derivative, the EE value, (for ultimate loads). EE values that are greater than two standard deviations above the mean for a certain quantity of interest are counted as significant. This was added as an equation instead of just written in the text to make it stand out more to the reader. The number of significant EE values coming from perturbations in a certain parameter are shown in Figures 11-14. A parameter with more associated significant EE values has a higher sensitivity. Figures 11-14 also show which quantity of interest the significant EE value is for.

Equation 5. added at the end of Section 2.4: "This is shown in Eq. (5), where SEE is a significant EE value, $\sigma$ is the standard deviation of EE values for a QOI, and $\mu$ is the mean EE value for a QOI. SEE > 2$\sigma$ + $\mu$"

*L316-318 as I find this part quite intricate for a fist time reader. If possible, an equation to reference when computing the "significant Ultimate Events" would go a long way here.*

Equation 5 mentioned in the previous comment is now referenced here in the text.

Added to Section 5 Paragraph 2: "All simulations were postprocessed for the relevant ultimate or fatigue EE values, and significant EE events were noted following Eq. (5)."

*L337: Possibly due to the uncertainty highlighted in my previous comment, what is the need for normalization?*

The ultimate load sensitivities were assessed across all three ultimate load cases with the appropriate safety factors. This covers the most sensitive input parameters for general extreme idling conditions. If someone is particularly interested in only one of the load cases, the breakdown mentioned here and shown in Figures 13 and 14, gives the sensitivities for the load cases individually. This breakdown can also provide insights into the causes of the sensitivities based on the differences in the load cases.

N/A

*L363-L382: If, as appears, the directionality of wind and waves matter, rather than the misalignment, then the absolute orientation of the platform also matters. This would make it quite hard to define "general" load cases, which has been the industry-standard approach up to this point. Perhaps a comment on this aspect could be added here or in the conclusions.*

The authors agree with this thought and recommend that future work looks at this specific sensitivity. The separation of misalignment and directionality was not considered until after finishing running the simulations, so only the one additional case shown in Figure 17 is available for analysis. From this case, it appears that the wave orientation relative to the platform may be equally or more important than the wave orientation relative to the wind direction, for this platform in these load conditions.
N/A

*Conclusions: It is particularly interesting that directionality parameters seem to have a large influence on quantities which should compensate for this by considering the sum of X and Y moments, such as the total tower base bending moment. An explanation on why this may be would be nice to see in the conclusions.*
The significance of directionality was larger than anticipated, and as the referee mentioned, not limited to loads in a specific direction. This sensitivity was found to be much larger in these extreme condition idling rotor load cases, compared to previous work with an operational turbine. In the idling condition, small asymmetry and misalignment can have large impacts. This shows up in the rotor loading with yaw misalignment, and in platform loading and motions with direction dependent hydrodynamic loads and mooring system properties. The second portion of this is likely platform and mooring system dependent.
Addition to Section 8 Paragraph 3: "Without the mean thrust and damping of an operating turbine, small asymmetries and misalignments can have significant impacts on loads. Changes to platform hydrodynamic loading and mooring system properties with direction are dependent on the specific device."

**Referee 2**
Comments:

*The manuscript proposes a sensitivity analysis of a multitude of environmental and turbine related input parameters to certain quantities of interest (QOI). Both ultimate and fatigue loads are considered. While similar studies have been carried out in the past (including by some of the authors), the scientific significance lies in the focus on extreme idling conditions (IEC load cases 6.1-6.5) rather than power production load cases. The study is carried out on the IEA15MW RWT atop the U Maine VolturnUS-S platform that is investigated for two sites of varying depth that require different mooring configurations (taut and semi-taut).*
*An elementary effects (EE) method is used to identify the parameters to which a given QOI is most sensitive. This required a large number of OpenFAST runs to be performed and analyzed in order to assign a sensitivity to a perturbation rather than a random combination of met-ocean conditions. The scientific approach and methods are valid and described extensively.*
*The manuscript is very well written and structured. The content of the work is presented clearly and reflects the title well.*
***Key outcomes can be summarized as:***
- *Ultimate load EE are generally sensitive to load directionality and thus to variations in inputs such as yaw, wave misalignment and current misalignment in idling extreme cases.*
- *The mooring configuration plays an outsized role as variations in the polyester length drives ultimate events in various QOI for the taut mooring system, while variations in this parameter are not that essential for the semi-taut configuration. Maximum wave heights are not essential.*
- *Fatigue load EE are also driven by directionality; however, in contrast to ultimate load EE, they are sensitive to wave height and periods as well.*

The authors agree with the highlighted key outcomes.
N/A

*L 61: This sentence seems seams somewhat detached from the rest- of the paragraph and might be better suited for the introduction.*
This sentence was meant to highlight the relevance of the selected turbine to the current offshore industry, and comment on the motivation for the creation of the reference design.
N/A

*L 95: While the decision to exclude dynamic stall models in these idling conditions is current industry standard and therefore reasonable, a lack of a dynamic stall model in idling conditions has been shown to also lead to non-physical instabilities (doi:10.1088/1742-6596/2626/1/012026).It would be valuable to hear the authors opinion on this.*
The authors agree with the importance of dynamic stall and deep stall in the load cases studied in this project. The choice was made to follow industry standard, since the currently available models are not well suited to the conditions of these extreme idling events. Future work should include improved models to eliminate the discussed unphysical instabilities. The paper mentioned (Bangga et al., 2023) discusses improved predictions with the IAG dynamic stall model; there are currently plans to incorporate this model into AeroDyn in 2026, which could allow for a valuable comparison.
End of Section 2.3 Paragraph 2: "This is the current industry standard for these load cases. Future work should be done to improve dynamic stall models for the relevant conditions; the IAG dynamic stall, for example, has been shown to predict more physically accurate loads (Bangga et al., 2023)."

*L 105: Mentioning the wave stretching method used in this study seems relevant as various studies in the past have shown its significance on non-linear excitation.*
Vertical stretching was applied, and this is now added to the text.
Addition to Section 2.3 Paragraph 4: "Vertical wave stretching was applied up to the elevation of the instantaneous free surface."

*L 176: $Twist_{root}$ is somewhat misleading. I believe what is described here is a pitch error/pitch misalignment*
While commented on in the text, it would be clearer to change the parameter name, and the variable has been renamed "Blade pitch error - $\Theta_{blade}$".
Change to Section 3 Paragraph 4: "$\Theta_{blade}$ is an error in the blade pitch, and the relative difference between $Twist_{tip}$ and $\Theta_{blade}$ is some error in geometric twist."
Input parameter name changed in Tables 2, 3, 4, B1, B2, B3, and B4 and Figures 11, 12, 13, 14, 15, 16, 18, 19, 20, 21, C1, C2, D1, and D2.

*Figs 4 – 8 would benefit from some more analysis. For example, the 500yr wind speed distribution of Humboldt Bay looks a bit odd and it is not clear to the reader where the lack of bins between 35 and 39 m/s comes from.*
A block size of 1 year was used due to the seasonality of the data. This results in a limited number of data points used in the bootstrapping resampling technique. The available 1-year maximum wind speed data from Humboldt Bay included two years with a larger value than the main cluster of data. If these points are not included in a resample, the upper tail of the distribution is much shorter, resulting in lower extreme

values. This effect is amplified for larger return periods further out on the GEV distribution. This doesn't necessarily indicate a bimodal distribution in the actual data, but indicates larger uncertainty in the extreme value. It was decided that it is best to include the outliers in the data, and account for the added uncertainty that they introduce.

Addition to Section 3.1.1 Paragraph 3: "The full set of yearly maximum wind speeds in the Humboldt Bay data had two data points with a larger value than the main cluster. When these years were not included in a resample, the upper tail of the GEV distribution is much shorter, resulting in a lower extreme value. This binary effect leads to a bimodal shape in the bootstrap distribution, that is more pronounced for larger return periods. This can occur because of a limited data set or because of outliers, both resulting in a larger uncertainty in the extreme value."

*L 290: Are the $V_{current}$ measurements above 3 m/s interpreted as a measurement error? This parameter already (at the capped value) shows a strong ultimate EE sensitivity which presumably would increase with a larger range of variation*

The current speed distribution in the Gulf of Maine data was more spread than in the Humboldt Bay data, leading to a larger bootstrapped uncertainty. There were no current speed measurements larger than 1.5 m/s, but the GEV distribution had a relatively long upper tail. Values larger than 3.0 m/s were deemed non-physical by expert opinion and the 500-yr value was capped accordingly. As the referee mentions, even with the capped range, current speed leads to significant sensitivities, highlighting the importance of understanding its uncertainty.

N/A

*L 377: Analyzing the separate influence of wave-only heading change and collective heading change is referred to as future work. A reference to this could be added in the conclusion*

A comment on the need for this parameter in future work has been added in the conclusion section.

Sentence added to Section 8 Paragraph 3 (second sentence here): "There is some ambiguity as to whether the relative misalignment of the waves to the wind or the waves to the platform is more important, but it appears that both angles matter. Future studies should include the relative angle between the waves and the platform as a variable input parameter."

*Sec. 6 Seed Convergence & Sec. 7 Starting Point convergence: These sections could be combined into a single section. It would also be sufficient to demonstrate convergence on a single site, if the manuscript is to be slightly shortened.*

While the specific findings of this work are of relevant interest, an important goal of the paper is to explain the methodology for others to perform similar studies with their specific system and location. For this reason, the authors would choose to keep the description of the seed and starting point convergence. If there is a need to shorten the manuscript, this advice will taken as a place to cut from, or perhaps moved to an appendix.

N/A